

**On the use of Weather Regimes to forecast meteorological drought over**

**Europe**

Christophe Lavaysse[*]

*European Commission, Joint Research Centre (JRC), 21027 Ispra (VA), Italy*

*Univ. Grenoble Alpes, CNRS, IRD, G-INP, IGE, F-38000 Grenoble, France*

Jürgen Vogt, Andrea Toreti

*European Commission, Joint Research Centre (JRC), 21027 Ispra (VA), Italy*

Marco L. Carrera

*Environment and Climate Change Canada, Dorval, QC, Canada*

Florian Pappenberger

*ECMWF, Reading, United Kingdom*

[*]*Corresponding author address:* European Commission, Joint Research Centre (JRC), 21027 Ispra

(VA), Italy

E-mail: christophe.lavaysse@ird.fr



## ABSTRACT

An early warning system for drought events can provide valuable information for decision makers dealing with water resources management and international aid. However, predicting such extreme events is still a big challenge. In this study, we compare two approaches for drought predictions based, respectively, on forecasted precipitation derived from the extended ENSemble system of the ECMWF, and on forecasted Monthly Occurrence Anomaly of Weather Regimes (MOAWRs) also derived from the ECMWF model.

Results show that the MOAWRs approach outperforms the one based on forecasted precipitation in winter in the northern and eastern parts of the European continent, where more than 65% of droughts are detected one month in advance. While, the approach based on forecasted precipitation achieves better performance in predicting drought events in central and eastern Europe in both spring and summer, when the local atmospheric forcing could be the key driver of the precipitation. Sensitivity tests also reveal the challenges in predicting small-scales and onset drought events at longer lead times.

Finally, in most of the cases, the ENSemble system of the ECMWF successfully represents the observed large scale atmospheric patterns, depicted by the MOAWRs, associated with drought events over Europe.



## 1. Introduction

Developing a robust early warning system for drought events is a key challenge for modelers and forecasters. The timescale of these events (generally from one to several months) requires accurate forecasts with long lead times. Due to the uncertainties of the models, the chaotic nature of the atmospheric circulation and the errors in the initial conditions, the reliability of precipitation forecasts is close to climatology beyond a 9-day lead time (Haiden et al. 2017). In a recent study (Lavaysse et al. 2015), it has been shown that about 40% of the meteorological droughts, defined by an anomaly of the standardized precipitation index (SPI), can be detected one month in advance by using the forecasted precipitation provided by the ECMWF extended ensemble. These forecasts might be improved by using post-processing techniques or predictors that are better simulated by atmospheric models. Most of the large-scale variability of rainfall and drought events is generated by specific large-scale circulation patterns.

The concept of Weather Regimes (WRs) was first introduced in the early 1950s, on the assumption that the atmosphere evolves between a finite number of large-scale circulation states. It is based on recurrent, persistent and/or quasi stationary states of the atmosphere (Michelangeli et al. 1995; Stephenson et al. 2004). They are well known to play an important role in creating large-scale conditions that either favor or inhibit precipitation in Europe, especially extreme events (Boé 2013; Guérémy et al. 2012; Yiou and Cattiaux 2013; Cattiaux et al. 2010; Toreti et al. 2010). These impacts can be observed in both winter and summer, and for different meteorological fields such as wind gusts and temperature extremes (Pfahl 2014). In Europe, WRs are highly teleconnected to temperature anomalies at the surface and precipitation (Plaut and Simonnet 2001; Yiou and Nogaj 2004) and well identifiable spatial patterns of these two variables are associated with each regime. For instance, the positive North Atlantic Oscillation phases (NAO+) in winter are teleconnected


⁵⁶ to above-normal temperature and precipitation over Northern Europe and below-normal precipita-

⁵⁷ tion over Southern and Central Europe (Wanner et al. 2001; Hurrell et al. 2003). Opposite results

⁵⁸ of surface temperature and precipitation anomalies are generally observed during negative NAO

⁵⁹ phases (NAO-). The WRs also have an impact on extreme events. The NAO+ regime favors heavy

⁶⁰ precipitation in northern Europe and periods of drought in the Mediterranean area. The Blocking

⁶¹ regime determines the occurrence of dry periods in large parts of southern Scandinavia and central

⁶² Europe (Yiou and Nogaj 2004). The use of WRs is also interesting since their occurrence and

⁶³ variability are connected to SST anomalies (Zampieri et al. 2017; Peings and Magnusdottir 2014;

⁶⁴ Häkkinen et al. 2011) and thus somehow takes implicitly into account the Atlantic ocean influence.

⁶⁵ The practical interest in classifying large-scale geopotential anomalies into a few pre-defined pat-

⁶⁶ terns relies on the fact that local weather conditions depend on large-scale atmospheric flows. If

⁶⁷ WRs can be better represented and forecasted by general circulation models (GCMs), they would

⁶⁸ provide additional information for local weather anomalies via statistical downscaling techniques,

⁶⁹ which derives teleconnections in between large scale geopotential anomalies (e.g., the WRs) and

⁷⁰ local weather phenomena (e.g., precipitation anomalies). Since geopotential and temperature

⁷¹ fields are generally better forecasted than precipitation in Numerical Weather Prediction systems

⁷² (Vitart 2014), we here analyze the benefit of using WR occurrences as predictor of drought occur-

⁷³ rence. A recent study carried out on the large scale forcing of long-term droughts has highlighted

⁷⁴ its potential interest for Europe (Kingston et al. 2015).

⁷⁵ Therefore, the objective of this study is to analyze the potential benefit of using atmospheric pre-

⁷⁶ dictors, and more specifically the WR occurrences, to forecast meteorological droughts in Europe.

⁷⁷ The paper is organized into six sections. The datasets and the methods are presented in section 2,

⁷⁸ the different forecast methods in section 3. The comparison of predictability scores obtained by


using precipitation and MOAWRs is provided in section 4. The sources of uncertainties are then

discussed in section 5 and the main conclusions are drawn in section 6.

## 2. Data and methods

*a. Datasets*

The observed daily cumulated precipitation data are retrieved from the Climate Assessment &

Dataset (ECA&D) and the ENSEMBLES gridded dataset (E-OBS) version 12, which provides ob-

servational daily station-based precipitation and temperature data on regular grids (Haylock et al.

2008). While the full E-OBS resolution is 0.25 degrees, data have been here upscaled by averag-

ing to 1 degree due to the specific focus on large-scale drought with significant socio-economic

impacts. E-OBS data are available from 1950 to the present.

Atmospheric predictors are identified by using the geopotential height at 500 hPa. The daily

geopotential is derived from the ERA Interim reanalysis (ERAI, Dee et al. 2011) with a spatial

resolution of 1.125 degrees covering the period from 1979 to the present.

The observed precipitation and the WR-based forecasts are compared to the forecast of precip-

itation taken from the extended ensemble system of the ECMWF (ENS hereafter, Molteni et al.

1996). The ENS is the latest version of the ECMWF ensemble model. In 2012, it was extended

once a week to a 32-day lead time and the horizontal resolution varies from Tl639 ( 32 km) from

t+0 to t+10days to Tl319 ( 64 km) from t+11 to t+32days. All of these datasets have been re-

gridded onto a regular grid of 1-degree resolution based on an averaging up-scaling method. The

rationale of using coarser resolution is i) to detect and so focus on larger scale precipitation deficits

and ii) to take into account spatial bias in the model that could detect the right precipitation signal

but with a slight spatial phasing error. The ENS is composed of one unperturbed member and





50 perturbed members, distinguished by different initial conditions and representations of model uncertainties. In addition to these forecasts, the ECMWF produces hindcasts that are launched the same date of the ENS for the last 20 years with five members only. To compute the forecasted WRs, the daily geopotential fields at 500 hPa for each member are extracted from the ENS system in both the hindcast and the forecast periods.

In order to build the baseline (following a normalization technique) and to have a time series long enough to calculate the scores, 21 years of hindcasts (November 1992 to November 2013) and the forecasts (November 2012 to November 2014) are used. To be coherent, the datasets of observed precipitation and the WR calculations (from ERAI) are restricted to the same period.

*b. Weather regimes*

The WR classification (i.e. definition of the WR patterns) is done, exclusively using ERAI, by a K-means method nested within a genetic algorithm to avoid dependence on the initial conditions and the trap of the local minima (Toreti et al. 2010). The four meteorological seasons are treated independently: winter (December to February), spring (March to May), summer (June to August) and autumn (September to November) but to avoid inconsistency moving from one season to another, each season is extended by adding the last month of the previous season. This method of classification has been extensively used (Michelangeli et al. 1995; Santos et al. 2005; Robertson and Ghil 1999), but because in this study the WR classification needs to fit with specific requests (20-year moving period of the hindcast, the four seasons), it is important to regenerate this classification. Nevertheless, the patterns of the geopotential anomalies (shown in Fig. 2 in supp. material ) are strongly similar to those obtained in previous studies mentioned earlier. The choice of using only the ERAI classification and not ENS is justified by: i) looking at previous studies that have shown the relatively similar behavior of ERAI and ENS forecasts (Ferranti et al. 2015); ii) consid-




ering that this choice avoids inconsistency (or the impossibility to derive a coherent classification)

due to the continuous evolution of the ENS model. Four WRs are identified in winter and spring,

while three WRs are detected in summer and autumn (see Fig. 2 in the supplementary material).

The number of WRs is estimated by following Toreti et al. (2010) and depends both on the period

(here from 1992 to 2013) and the region (North Atlantic) studied. This is why the number of

summer WRs is different with respect to, e.g., Cassou et al. (2005).

Then, an assignation procedure is run to identify The closest WR to a given daily geopotential

anomaly of ERAI and a given ENS-member. To this aim, the method proposed by Ferranti et al.

(2015) is here applied. Namely, a pattern matching algorithm based on the minimum distance

from the previously identified centroids is used to assign each day and individual forecast member

to the closest weather regime (previously identified by using exclusively ERAI). The climatology

of the forecasted WRs is then calculated by summing the daily classification of each WR for all

the members and all the days inside a 30-day window. The same climatology is derived by using

ERAI. The Monthly Occurrence Anomaly of WRs (MOAWRs) is then calculated with respect to

the climatological occurrences based on the hindcast period (1992-2013) and obtained by using

independently both ERAI and ENS.

To potentially increase the signal emerging from the teleconnection between MOAWRs and

precipitation anomalies, different combinations (additions and subtractions of WR occurrences)

of two WRs are tested. This could be useful when two WRs have the same or opposite impacts

on precipitation. For example, in the case of two regimes WRa and WRb that are associated with,

respectively, dry and wet conditions over a certain region, the occurrence difference of the two

WRs could be more linked to the drought events over that region (this example will be discussed

later in the document). In total, a set of 6 to 12 combinations (see the list in Table 1) is tested when

3 or 4 WRs (depending the season) are detected, respectively.



*c. Drought metrics*

As suggested by the World Meteorological Organization, the Standardized Precipitation Index (SPI) is one of the most relevant indicators providing a clear and robust characterization of precipitation deficiencies and it is a good proxy for assessing meteorological droughts. The SPI calculation is relatively simple and it is performed independently at each grid point of the domain. This method is robust and has the advantage of being flexible in time, for the accumulation period studied, and in space, for the resolutions used. It also provides an unbiased product, which is important for comparing datasets from observations and model simulations. In the SPI calculation, a gamma distribution is first fitted to precipitation data and then transformed into a standard normal distribution (McKee et al. 1993, 1995; Svoboda et al. 2012). The choice of the statistical distribution has been verified in Lavaysse et al. (2015) and shown that this assumption is valid over a large proportion of Europe. Nevertheless, over the driest regions and in summer, some grid points (mainly in Spain and South Italy) the significant tests are not verified. Both the observed and modeled daily precipitation values are accumulated over a period of 30 days (i.e. we use the SPI-1, where 1 refers to the accumulation period of one month). The choice of analyzing relatively short meteorological droughts is based on two main constraints: i) a technical one connected to the limitation of the extended ENS that provides forecasts up to 33 days in the version here analyzed, and ii) the chaotic nature of the atmosphere that limits the predictability of precipitation and geopotential forecasts after several weeks. This relative short term drought information is also relevant for users and decision makers since it provides valuable information about the onset, continuation or end of longer droughts.

Based on this approach, both observed and forecasted SPI-1 values are calculated for the period 1992-2013. Here, a meteorological drought is defined as having SPI-1 values less than -1.




According to the normal distribution of the SPI, this threshold corresponds to about 17.5% of the

driest events. Based on Lavaysse et al. (2015), the most reliable method for producing a dichoto-

mous forecast of drought from probabilistic forecasts of precipitation, and more specifically from

the extended ENS of the ECMWF, is to predict a drought as soon as more than 30% of the ENS

members are associated with a drought forecast (i.e. SPI-1$< -1$).

*d. Validation tools*

To assess the forecasts of drought events, traditional scores for dichotomous products are ap-

plied. These scores make use of the contingency table (Table 2) that shows the types of agreement

of observed and forecasted variables.

The percentage of observed events that had been correctly forecasted are provided by the

Probability Of Detection score (POD) whereas the percentage of events that had been forecasted

but did not occur are indicated by the False Alarm Rate (FAR). Finally, to take into account the

hits, misses and false alarms and to neglect the correct negative forecasts that will dump the

scores for rare events, the Gilbert Skill Score (GSS, Jolliffe and Stephenson 2003) is used. For

rare events, such as droughts, it is more relevant to use this score than the Peirce's skill score for

instance. The GSS indicates how well the forecasted droughts correspond to the observed ones.

This skill score is compared to the score obtained by the climatology. It is calculated as follows:

$$GSS = \frac{(hits - hits_c)}{(hits + misses + false\ alarms - hits_c)} \tag{1}$$

where $hits_c = \frac{(hits+misses)(hits+false\ alarms)}{total}$. Based on these equations, a perfect forecast achieves a

score equal to 1, while a score equal to 0 is assigned to the climatology (i.e. no forecast skill). All

these scores are calculated independently for each season.





## 3. Configuration of the drought forecasts

To forecast droughts using the MOAWRs approach, 3 steps are needed (see also Fig. 1 in the supplementary material): 1) the WR classification (detailed previously and exclusively done with ERAI); 2) the daily WR attribution, to determine which is the closest WR classified previously for a forecasted (or reanalysis) geopotential anomaly for each day and member; 3) the predictor assignation, to determine which WR is the best predictor of droughts for each grid point. This last step is based on a best correlation criterion and leads to coherent picture (Fig. 1), highlighting the large-scale impacts of the WRs. It is interesting to note that following this approach, the large majority of SPI-1 in Europe is associated with a combination of WRs. The correlation values used in the last step show significant spatial differences (Fig. 2). Throughout the year, they are generally higher teleconnections in northern Europe than in southern Europe. This could explain the larger variability of MOAWR predictors over Central than North Western Europe. There is also a strong seasonal difference. In winter, the mean correlation is about 0.55 whereas it is about 0.28 in summer. The origin of precipitation, which is more synoptically-driven in winter and more local in summer, can explain these results.

Once the attribution of WRs and the predictor assignation are done, the potential benefits is assessed to analyze the limitations of using these predictors. To do so, 5 different drought forecasting approaches (that are differentiated by the methodologies employed for the three steps listed previously) are used. The detailed configuration of these methodologies is provided in the supplementary material, while a brief overview is here reported.





The first method of drought forecasting is based on forecasted precipitation (Lavaysse et al.

2015). The skill scores of this forecasting approach are here used as benchmark (see Table 6

for the list of these characteristics). The second approach, called 'idealized forecast', uses exclu-

sively the MOAWRs derived from ERAI and does not take into account the uncertainties related

to the forecasts of WRs. The third forecasting method, called operational forecast, is based on the

MOAWRs derived from the forecasted WRs of the ensemble model (nevertheless, ERAI is used

for the WR classification to avoid potential problem when the version of the operational model

changes). The fourth forecasting scheme, called optimized forecast, is similar to the previous one

but uses ENS for the WRs assignation. This methodology tends to optimize the forecasts by cor-

recting some bias in the forecasted MOAWRs. Finally the fifth forecasting method, called process

forecast, is similar to the others except for the use of modelled precipitation. This last approach

allows to analyze the modelled teleconnection between MOAWRs and SPI; therefore, it can be

used to investigate the skill of the model in representing observed processes.

## 4. Results

*a. Skill scores*

The skill scores of the forecasted precipitation, also called and presented in the previous section

as Reference and used as benchmark, provided by ENS and derived from Lavaysse et al. (2015),

where a drought is forecasted when at least 30% of members forecasts $SPI < -1$, are here shown.

The best achieved performance (for winter in Central Europe) shows how slightly more than 40%

of the observed drought events are correctly predicted with a 30-day lead time (Fig. 3a, d, g, j) with

about 60% of false alarms (Fig. 3b, e, h, k). For both POD and FAR, the spatial variability is small

(standard deviation lower than 0.2) especially during spring and fall. In winter, an improvement




²³⁴ can be noticed in Germany, Poland, Spain and Norway; whereas in summer, drought seems to be

²³⁵ more predictable in eastern Europe. When the POD and the FAR are combined in the integrated

²³⁶ GSS (Fig. 3c, f, i, l), higher seasonal and spatial differences appear. Overall, the score reaches

²³⁷ up to 0.3 in winter, especially in northern Germany [50deg.N, 10deg.E]; while the worst value

²³⁸ is reached in spring and summer, especially in western Europe (France, Belgium). Due to the

²³⁹ impacts of the local forcing on precipitation, the drought forecasts based on large-scale predictors

²⁴⁰ are better in continental than in coastal regions (more details in Lavaysse et al. (2015)).

²⁴¹  The forecasts based on MOAWRs can now be assessed with respect to the precipitation-based

²⁴² forecast (SPI-1). In order to detect the same number of drought events when using these predictors

²⁴³ and the precipitation, the threshold of the MOAWRs is chosen equal to 0.176 (0.824 for negative

²⁴⁴ correlations). The POD, FAR and GSS for the four seasons are shown in Fig. 4 using 20 years with

²⁴⁵ a leave-one-out technique, which is a cross-validation method for small samples sizes enabling to

²⁴⁶ validate results by simply partitioning the series into a training and a test part. To highlight the

²⁴⁷ benefits of this method, the anomaly with respect to the reference forecast (i.e. Fig. 3) are plotted.

²⁴⁸ The forecast based on MOAWRs is more spatially variable. As for winter in the northern part

²⁴⁹ of Europe, this forecast is significantly better in terms of both POD and GSS; whereas in central

²⁵⁰ Europe the forecast based on precipitation is more reliable. Despite the fact that the patterns are

²⁵¹ less homogeneous for the other seasons, some positive impacts of this forecasting approach appear,

²⁵² e.g. in: northern Russia in spring, western Europe in summer, central Europe during fall.

²⁵³  These results are consistent with the intensity of the teleconnection measured during the assig-

²⁵⁴ nation procedure between the SPI and the WRs (Fig. 2a) and highlight the regions where the

²⁵⁵ large-scale atmospheric patterns associated with the WRs could better explain strong precipitation

²⁵⁶ deficits when compared to local drivers (orography, soil moisture, coastline).





### *b. Intensity and initial conditions*

To better understand the potential performance of the approach, sensitivity tests are conducted. In the previous section, the SPI-1 intensity threshold to define a drought was fixed to -1. The previously used skill scores are here derived for SPI lower than -1.5 and -2 (respectively $\sim 7\%$ and $\sim 2.5\%$ of the most extreme cases). A second sensitivity test is done on the initial conditions influencing all results but also bringing useful information on drought onset and persistence. Most of the studies on drought focuses on 3-month (or longer) cumulated precipitation that could have more severe impacts on, e.g., agriculture and water resources. Due to the unpredictable nature of the weather and the limitation of the lead time of the ENS model, the assessment of drought forecasting is limited to 1-month lead time in our study. Nevertheless, the information of the two previous months (observed SPI-2 with a threshold defined as -1) is taken into account to measure the impacts of these initial conditions and the ability to forecast drought persistence/onset.

In Fig. 5 the GSS scores in winter for all the domain shown in the previous Figures are synthesized by using boxplots. The results shown in Fig. 3a and Fig. 4a are represented by the black boxplots plotted for SPI $< -1$ in Fig. 5a and b respectively. Overall, the predictability decreases with the drought intensity. In winter, dry initial conditions generate a favorable environment to better forecast droughts. In other words, the persistence of drought is better predicted than the onset. Finally, the last main result concerns the improvement of the forecasts based on MOAWRs. For the SPI lower than -1, all GSS values shown in Fig. 5 are quite close. But, as also highlighted by Fig. 4, there is a larger spatial variability with the MOAWRs approach. For more intense droughts, there is a global and significant improvement with MOAWRs. Indeed for drought intensities with SPI lower than -2, the median of the GSS scores goes up from close to 0 (using the

precipitation-based method) to 0.05 (using the MOAWRs). This result is mainly explained by the

better predictability of the drought onset (right blue boxes in Fig. 5a and b).

The same sensitivity tests are conducted for the others seasons (not shown), and the decrease

of predictability with increasing drought intensity is found for all of them. Nevertheless, the

conclusions on the role of the initial conditions depend on the season. For instance in summer,

drought onsets are slightly better predicted than drought persistence. The reason could be the

higher temporal variability of the monthly precipitation deficits in summer than in winter due to

the larger impact of local forcings. Finally in all the seasons, the use of atmospheric predictors

leads to better performance when looking at the most extreme events (SPI $< -2$).

## 5. Sources of uncertainty

To better discuss and understand the previous results and their uncertainties, additional tests

are here reported. The main objective is to quantify the contribution of the uncertainties in WR

predictions and the teleconnection between the SPI and the WRs.

*a. Validation of the WR forecasts*

The first question to address is about the quality of the forecasts of MOAWRs. The method

used is based on the total occurrence of each WR among all the members and the entire lead time

(5 members$*$ 30-day LT). The anomalies are then calculated in relation to the climatology of the

forecasts. These anomalies are divided by the number of ensemble members to create comparable

results with the data provided by ERAI.

To validate the forecast of the WRs, first the comparison of the frequency of occurrence of each

daily WR is performed (Fig. 6). The WR-distributions as given by the forecasts are characterized

by a higher degree of similarity than the ones given by ERAI, with a peak of occurrence at around





5-8 days in winter (blue bars, Fig. 6). The same holds for the other seasons (not shown). The lower

spread of the forecasted WR occurrences, associated with reduced tails (i.e. reduced occurrences

for durations exceeding 20 days), could be explained by the underestimation of the long-term

blocking. A further examination of the temporal evolution of these occurrence anomalies suggests

that the distribution of forecasted drought occurrences (previously shown) could mainly explain

the overestimation of low occurrences in the observations (i.e., larger number of forecasted events

compared to observed ones with durations shorter than 5 days) and the underestimation of longer

duration events (i.e., lower forecasted than observed events with durations longer than 15 days, red

dotted lines in Fig. 7). Despite this behavior, the correlations appear significant with a maximum

of 0.65 for the WRa (significance with 90% of confidence at 0.58). These significant scores are

obtained in winter, while for the other seasons the correlations are lower (see Table 6). In summer,

they are not significant for two-thirds of the WRs.

*b. Strength of MOAWR-precipitation teleconnection*

According to the previous subsection, the WR forecast could be improved. Thus, it is important

to assess the limitation of the method using predictors and so assessing the strength of the MOAWR

and precipitation teleconnection. To this aim, the procedure 'idealized forecasts' of MOAWRs, e.g.

provided by ERAI without uncertainties on forecast, are compared to the forecast of precipitation

discussed and shown in Fig. 3.

The POD scores are strongly improved between seasons and regions (Fig. 8a, d, g, and j). These

results are strongly connected to the correlation values obtained and shown in Fig. 2 with the same

North-South and seasonal variabilities being observed. However, almost all the grid points show

a better POD with the WR predictors than with the precipitation-based forecasts. Up to 70% of

observed drought events are correctly detected in northern Europe during winter. This percentage


falls to about 17.5% in summer (i.e., the climatological value) in the southern part of the domain.

The results in terms of FAR are more variable depending on both the season and the region. On

average, there is a small decrease of the FAR. However, the GSS shows a clear and significant

improvement in the drought forecast when using the WR predictors. Compared to the scores using

real forecasts in Fig. 4, the bigger difference is more in terms of intensities than in the spatial

distribution. For instance, in winter a high improvement is observed in northern Europe (up to 0.2

against 0.1 for the real forecast over Scandinavia) whereas a decrease/close score is obtained in

central Europe. Based on this sensitive analysis, the teleconnection between the SPI-1<-1 and the

MOAWRs is strong enough to provide significant improvements of the prediction scores in most

of the regions. Nevertheless, this analysis also highlights the limitations of the methods used in

this study when and where the influence of the WR on drought is lower (i.e. Germany and Poland

in winter; eastern Europe in summer; southern Europe in fall).

*c. Modeled teleconnection*

Some additional tests are also conducted on the predictor assignation procedures (definition of

the best predictant for SPI-1 < -1 at each grid point) to see the impacts of using either ERAI

or ENS (the latter could potentially correct bias of the ENS, see the supplementary material for

more details). Due to the errors associated with the WR forecasts, the procedures using WRs

from ERAI or ENS provide different results (Fig. 9a, c compared to Fig. 1). The assignation pat-

terns done with the WRs provided by ERAI (Fig. 1) have less homogeneous large-scale structures

(i.e., more spatial variability) than those provided by the WRs forecasted by ENS (Fig. 9a and c)

showing a more complex observed than forecasted teleconnections. Nevertheless over continental

regions, there are some similarities between the assigned predictors obtained by using ENS and

ERAI (impact of WRs b, b-a, a, c-d), illustrating the relative good representation of the impacts



of specific WR on precipitation by ENS. The correlation between the WRs forecasted and the observed precipitation are then plotted (Fig. 9b). The correlation values, which can be compared to the correlation shown in Fig. 2a, are quite low as a result of the relatively low predictability of the WRs previously discussed and where the teleconnection between WRs and precipitation is the highest (i.e. in southern Norway and the northern part of the U.K.).

The last analysis is focused on the modeled teleconnection between the SPI and the WRs both provided by the ENS (Fig. 9c and 9d). It is remarkable the great similarities of the maps for the assignations and the observed teleconnection by using observations and realanysis (correlation values greater than 0.65, Fig. 1 and 2a). This is especially true over the U.K, Ireland, Scandinavia, Spain and north-western Russia. Some differences are observed in southern France and Italy where the model overestimates the large-scale forcing on precipitation (i.e. with stronger correlation with WRs than observed). This highlights the overall good representation of the processes linking large scale circulation and precipitation deficits by ENS. So the ENS model succeeds in capturing the impacts of the WR occurrence on the precipitation anomalies as shown in the observations over a large part of Europe. These results could suggest limitations in using such predictors as the lack of skill score could result from a failure in forecasting the large-scale atmospheric circulation (rather than from a misrepresentation of the physical processes).

## 6. Conclusion

In this study, a drought forecasting method based on large-scale atmospheric predictors is proposed in order to improve the early warning of atmospheric drought events. The method is based on the Monthly Occurrence Anomalies of Weather Regimes (MOAWRs) within a 30-day lead-time. The methodology used to select the predictors is based on a three step procedure. First, WRs (described by daily geopotential anomalies) are identified by using a Genetic K-means algorithm





for each season separately and for both ERAI and extended ENS forecasts. The climatological

occurrences are calculated for each WR. The identified three/four WRs (depending on the sea-

son) are combined (added or subtracted) with each other to enhance the potential signal of their

impacts. Second, the MOAWRs is used as a predictor of meteorological droughts at each grid

point. The predictor assignation procedure is based on the correlation between the MOAWRs and

the SPI-1. To select the best predictor, the MOAWR associated with the strongest absolute value

of correlation is selected. The last step involves the forecasting of the SPI-1 lower than -1. Two

approaches are derived and compared. The first one is based on the index developed by Lavaysse

et al. (2015) for drought events and derived from the forecasted precipitation provided by the ENS.

This represents a benchmark for the early warning of drought forecasting. At most around 40% of

drought events are detected one month in advance with 65% of false alarms. The second forecast-

ing approach is based on MOAWRs. In the northern and eastern parts of the European continent,

an improvement of the Gilbert Skill Score (GSS) is observed w.r.t. the precipitation-based one.

Nevertheless, this is balanced by other regions where the forecast skills is clearly lower (central

Europe in winter, eastern Europe in summer) than a precipitation based one. The origin of this

spatial and temporal variability in the skill scores is associated with the dynamic of precipitation.

In winter, precipitation is much more related to large-scale atmospheric forcing mainly captured

by the MOAWRs. On the contrary in summer, precipitation is more affected by local forcings

that could influence, for instance, the trajectory and the occurrence of convective systems. In this

study, this behavior is captured by the better correlation between MOAWRs and precipitation in

winter than in summer. The spatially variable skill scores is mainly controlled by the intensity

of the teleconnection between the MOAWRs and SPI-1. Due to the location of the geopotential

anomalies and the induced wind fields, or connected with the local effects that reduce the influ-

ence of the large scale forcing, the impacts of these MOAWRs on precipitation could be low, as



observed in winter over central Europe. According to these scores, the most reliable forecast could result from choosing the best method for each grid point independently. The influence of the initial conditions and the intensity of the drought highlight i) the losses of predictability with increasing drought intensity and ii) the better scores in predicting persistency rather than the onset of drought especially in winter. Also, the benefits of using the WRs to predict droughts appear to be more important when the most intense droughts (i.e. SPI < -2) are forecasted.

This study shows the importance of improving the prediction of the WR occurrences. The methodology applied here could be compared to more complex methodologies using clustering of the members to define the most probable scenario, or by taking into account the transition between WRs. Future work should also take into account the uncertainties in WR prediction. Recent studies (Matsueda and Palmer 2014, 2015) have shown that WR prediction is still a big challenge with regard to lead-times greater than 15 days. Some improvements could be also done by using a multi-model ensemble such as the one recently developed in the framework of the Seasonal to Sub-Seasonal (S2S) project (Vitart et al. 2016).

*Acknowledgments.*

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





## LIST OF TABLES




| Name | MOAWR predictor | WR for assignation | SPI for assignation |
|---|---|---|---|
| Reference | no predictor (Precip. forecast) | - | - |
| Idealized | ERAI | ERAI | Observed |
| Operational | ENS | ERAI | Observed |
| Optimized | ENS | ENS | Observed |
| Process | ENS | ENS | Forecasted (ENS) |

TABLE 1. Definition of the four sets of forecasts compared in that study. The differences are based on the use

of predictor or not, the use of reanalysed or forecasted WRs as predictor and for the assignation procedure and

on the use of observed or forecasted SPI during the assignation procedure.



| WR | A | B | C | D |
|----|---|---|---|---|
| A | WRa (#13) | a+b (#07) | a+c (#08) | a+d (#09)* |
| B | a-b (#01) | WRb (#14) | b+c (#10) | b+d (#11)* |
| C | a-c (#02) | b-c (#04) | WRc (#15) | c+d (#12)* |
| D | a-d (#03)* | b-d (#05)* | c-d (#06)* | WRd (#16)* |

TABLE 2. Definition of WRs and WR combinations. * indicate regimes that exist only in winter and spring.





|  |  | Event observed | |
|---|---|---|---|
|  |  | Yes | No |
| Event | Yes | hits | false alarms |
| Forecasted | No | misses | correct negative |

TABLE 3. Contingency table of dichotomous events illustrating the four types of classification between ob-

served and forecasted events.





| Season | WR | Correlation | Season | WR | Correlation |
|---|---|---|---|---|---|
| Winter | A | **0.65** | Spring | A | **0.48** |
| | B | **0.52** | | B | 0.43 |
| | C | **0.57** | | C | **0.47** |
| | D | **0.57** | | D | **0.62** |
| Summer | A | 0.45 | Autumn | A | **0.51** |
| | B | 0.42 | | B | **0.47** |
| | C | **0.47** | | C | **0.47** |

TABLE 4. Correlation values between the WR occurrence forecasted and observed for each WRs and the four

seasons. Values indicated in bold have a significance level above 0.9.





## LIST OF FIGURES





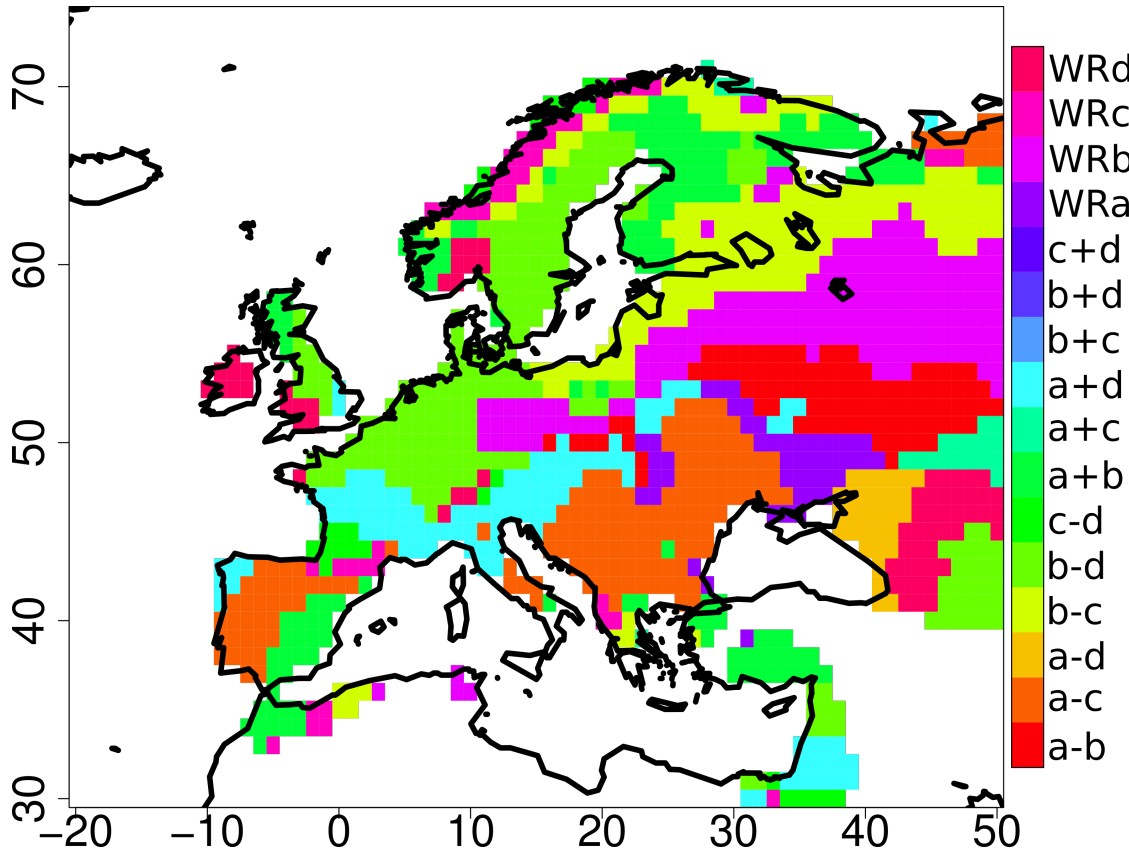

FIG. 1. Automatic attribution of the best predictors in winter based on the occurrence anomalies of WRs of
ERAI and the observed precipitation. The names of the predictors are indicated in the color scale.





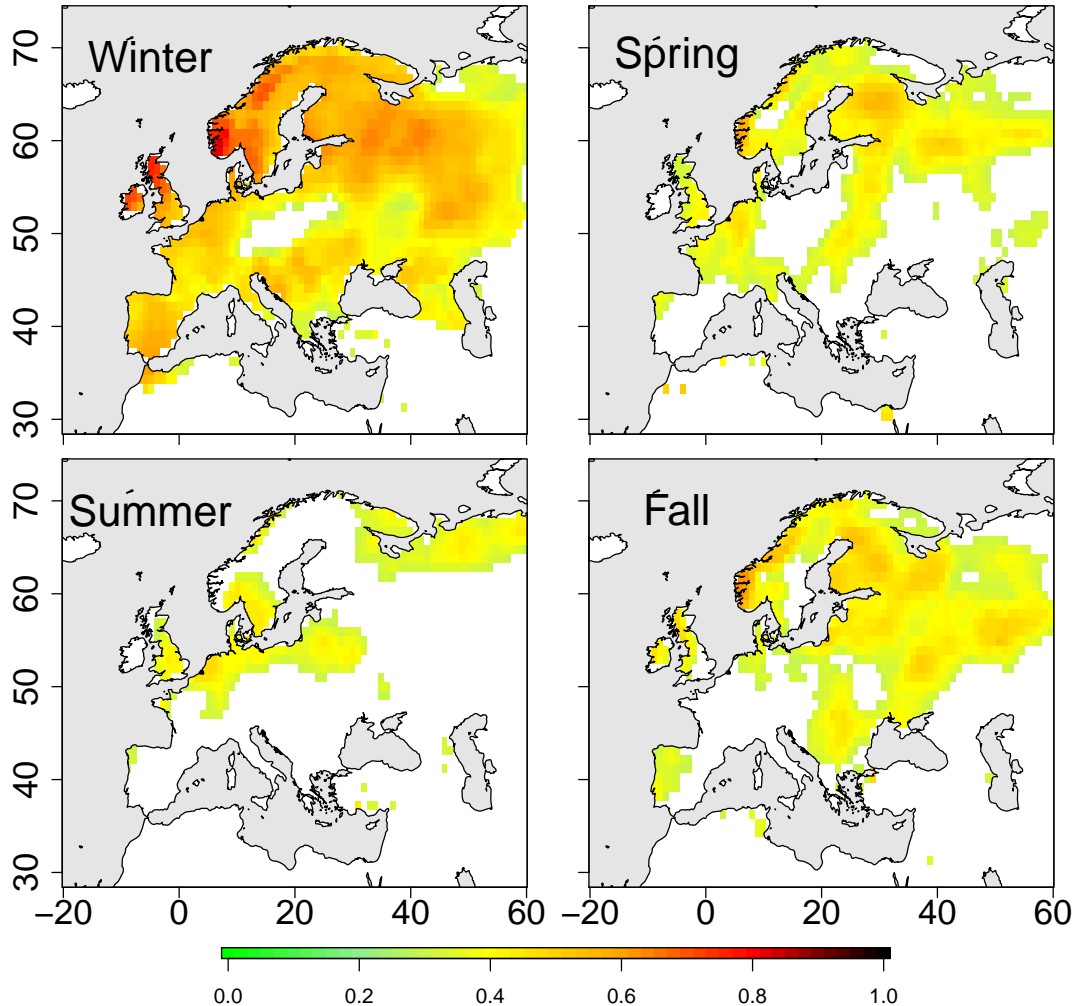

FIG. 2. Absolute values of temporal correlation between SPI-1 and MOAWR attributed from the 16 combina-

tions in winter (a), spring (b), summer (c) and autumn (d). Only values with a confidence level larger than 90%

are plotted.



FIG. 3. POD (left panels), FAR (centre panels) and GSS (right panels) scores of droughts prediction calculated using the forecasted precipitation. The scores are calculated for (from top to bottom) winter (first), spring (second), summer (third) and autumn (fourth line).



FIG. 4. Anomalies of POD (left panels), FAR (centre panels) and GSS*2 (right panels) scores of drought prediction calculated using the MOAWR in relation to the forecasted precipitation (see Fig. 7). The scores are calculated for (from top to bottom) winter (first), spring (second), summer (third) and autumn (fourth line). Improvement scores by using the predictors are indicated in green (inverse scale for FAR). Only difference with confidence interval larger than 90% are plotted.





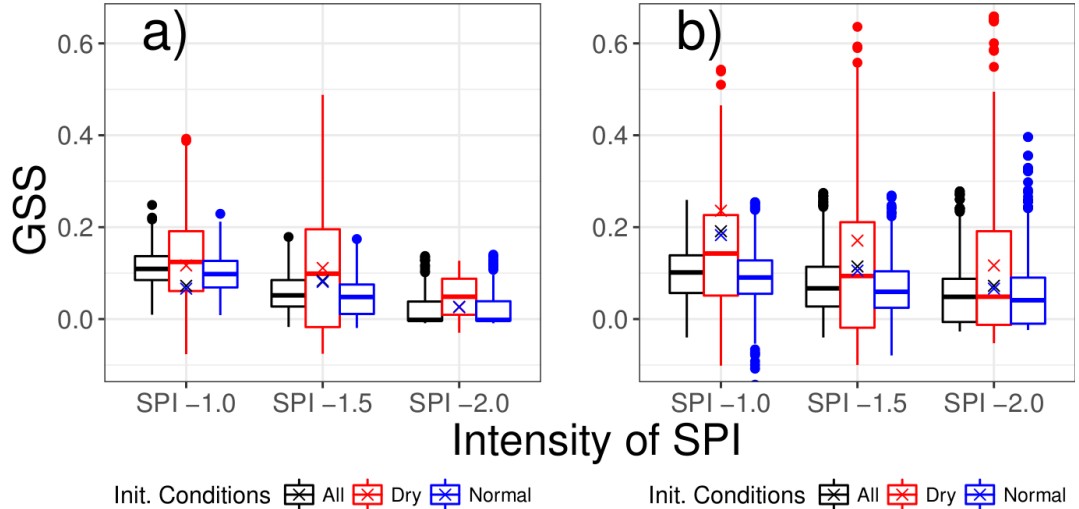

FIG. 5. Boxplot of the GSS scores in winter using the forecasted precipitation (a) and the MOAWRs (b). The scores are calculated over the entire domain and the boxes display the spatial variability. The scores are depending to the SPI intensities (-1, -1.5 and -2, x-axis) and the initial conditions defined by the previous SPI-2 conditions (see text for more details). Crosses indicate the scores but calculated by merging all the grid cells.





FIG. 6. Example of frequency distribution of WR occurrences (in days per 30-day windows) in winter for WR-A (a), WR-B (b), WR-C (c) and WR-D (d) using ERAI and ENS (red and blue bars respectively, purple when the two overlap).





FIG. 7. Scatter plots of the occurrence of the four winter WRs provided by ERAI (x axis) and provided by ENS (y axis). The linear least square regression are indicated with red dashed lines and the correspondent correlation on the top right of each panel.



FIG. 8. Anomalies of POD (left panels), FAR (middle panels) and GSS*2:w (right panels) of the drought
prediction based on idealized forecasts of drought (using ERAI) and on precipitation forecasts, in winter (first),
spring (second), summer (third) and fall (fourth line). Improvement scores by using the predictors are indicated
in green (inverse scale for FAR). Only difference with confidence interval larger than 90% are plotted.




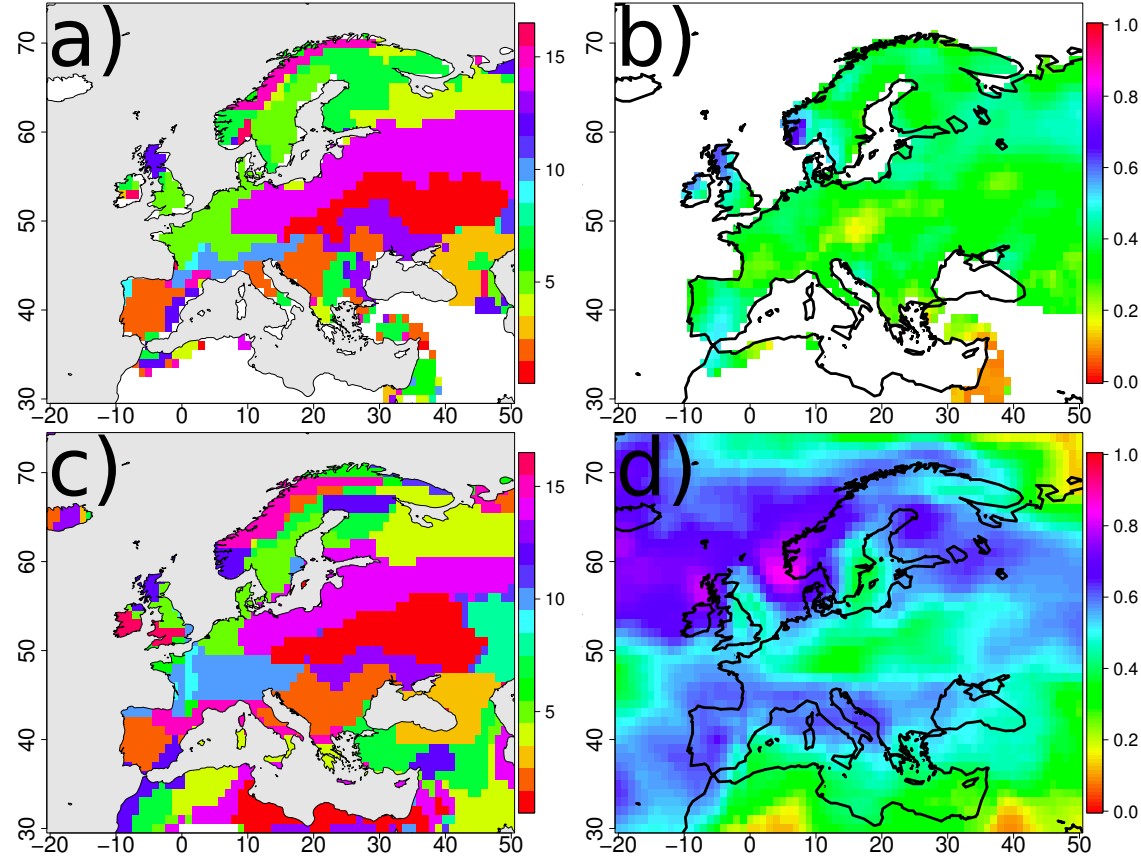

FIG. 9. Assigned winter WR (left panels) and associated correlation values (right panels) for 'Optimized Forecast' (top): predictors defined using MOAWRs from ENS and observed SPI-1 (a), correlation calculated between the forecasted MOAWRs and observed SPI-1 (b); and 'Process Forecast' (bottom): predictors defined using MOAWRs from ENS and forecasted SPI-1 (c), correlation calculated between the forecasted MOAWRs and the forecasted SPI-1 (d).