# Peer review of "On the use of Weather Regimes to forecast meteorological drought over Europe"

_Natural Hazards and Earth System Sciences, 2018_

## Referee Comment (RC1) · Anonymous Referee #1 · 2 Aug 2018

The authors present a novel approach to provide early warnings for drought events over Europe, based on ECMWF's ERA-Interim reanalysis and ENS forecast system. I really like their basic idea, as it combines a scientifically interesting result with a very practical and applications-oriented framework. Furthermore, the authors have clearly performed an extensive set of analyses in support of their results.

However, I think the manuscript falls short of publication requirements in its current form. In addition to some comments of a more scientific nature, my main concern is the poor form of the submission. The text is often grammatically flawed or very unclear; references to tables are incorrect (and in one case to a table that doesn't exist, at least in the version of the paper I reviewed), and some existing tables are never referenced. Overall the manuscript has a very unrefined feel.

[Figure]

I reiterate that I do find the motivation behind the study and the authors' approach of great interest, so I would encourage them to perform a very thorough review such that their results may be published in NHESS.

**Major Comments**

1. As stated above, I deem the manuscript to be in an advanced draft form rather than at a publication-ready stage. The issues range from simple typos/grammatical errors/incorrectly referenced tables/oversights (some examples here:

l. 25 "While, the"
l. 29 "and onset drought events"
l. 69 "teleconnections in between"
ll. 84-85 "observational daily station-based" This is a bit redundant. If something is station based it is very likely to also be observational.
ll. 96 Space between "10" and "days" and "32" and "days". Remove multiple spaces between "Tl319" and "64".
l. 97 "1-degree" no need for hyphen.
l. 102 "launched" –> "initialised"
l. 111 "done, exclusively"
l. 121 There is a lone parenthesis
l. 121 "previous studies mentioned earlier" –> "aforementioned studies".
l. 130 "identify The closest"
l. 146 "Table 1". Do the authors here mean Table 2? The real Table 1 actually does not seem to be referenced anywhere in the text.
l. 156 "datasets from observations" –> "observational datasets"
ll. 157-158 "The choice ... has bee verified in Lavaysse et al. (2015) and shown that this assumption"
ll. 159-160 "some grid points the significant tests are not verified"
l. 178 "Table 2" –> "Table 3"?
l. 183 "dump" –> "lower"

l.185 "Peirce" –> "Pierce"

l. 198 "leads to coherent picture"

ll. 201-202 "they are generally higher teleconnections"

ll. 207-208 "the potential beneifts is assessed"

l. 213 There is no table 6 in the version of the paper I have reviewed (indeed the tables stop at 4).

l. 218 "to avoid potential problem"

ll. 227-228 "also called and presented in the previous section as Reference"

l. 229 "forecasts" –> "forecast"

l. 247 "anomaly" –> "anomalies"

l. 261 "2.5l. 269 "for all the domain shown in the previous Figures" –> "for the whole domain shown in the previous figures"

l. 311 Again a reference to Table 6.

l. 316 "e.g." or "i.e."?

l. 328 "intensities" –> "magnitudes"?

l. 344 "showing a more complex observed than forecasted teleconnections"

l. 346 "the relative good representation"

l. 347 "The correlation between the WRs forecasted and the observed precipitation"

l. 385 "the dynamic of precipitation"

l. 390 "The skill scores is"

l. 405 "with regard to" –> "for"

l. 408 Acknowledgements are missing.

Tables 3 and 4 are never referred to in the text.

Fig. 8 caption "GSS*2:w"

The figures in the SI aren't prefaced by "S" which makes it hard to figure out whether the authors in the SI refer to the SI figures or those in the main text.

l. 72 in the SI: "?" in place of a reference.)

to some very unclear or contradictory passages that I would recommend the authors re-phrase (some examples here:

Interactive
comment

ll. 30-32 This is a bit confusing: if the MOAWRs are anomalies of occurrence how can they "depict" a large-scale atmospheric pattern?

ll. 48-52 The authors first discuss precipitation and then switch to wind gusts and temperature extremes without any apparent connection. How are the latter two fields relevant to the study?

ll. 202-203 Why speak of northern and southern Europe and then shift to central and north western Europe?

ll. 212-224 This passage is very important (the authors explain a key aspect of their approach) but also occasionally difficult to understand. For example, what does "uses ENS for the WRs assignation" mean? Do the authors mean that the WRs are defined using the ENS dataset? Similarly, what does "modelled precipitation" refer to? Do the authors mean ENS precipitation (ERA-Interim is a model too)? Linked to the above, the passage describing the forecasting methodologies in the SI (to which the authors point the reader) never explicitly mentions the "idealized" approach, although this is included in Fig. S1. I would suggest the authors simply include the full description of the methodologies in the main paper (the text in the SI is not that much longer than that already in the main paper), leaving the details of the attribution and MDA analysis in the SI. Again linked to the above, since the authors have given names to the different forecasting approaches, they should use them! For e.g. the caption in Fig. 4 never names the apporach it depicts. This is one example, but there are a number of other similar cases throughout the submission.

ll. 228 Actually, this is never called "Reference" in the previous section.

ll. 305-307 The authors first speak about "overestimation of low occurrences in the observations", which is confusing because observations (or rather reanalysis, if I understood what the paragraph talks about) here are taken as the ground truth and so cannot under or overestimate occurrences. Next they mention the "larger number of forecasted events compared to observed ones with durations shorter than 5 days". I take this to mean that the forecasts produce a larger number of short events than the reanalysis. However, from Figure 6 it seems that short events are more frequent in

ERA-Interim than in the ENS data. I may have misunderstood the whole passage, but in that case other readers may well have the same issue.

ll. 346-351 From this sentence I understand that the correlation values are low where the teleconnection with precipitation is the strongest, while Figure 9 (and logic) suggests the opposite.

ll. 361-363 Do the authors mean that the results clearly point to the ability in forecasting the large-scale atmospheric circulation as a factor limiting the skill of this approach?

ll. 391-393 I think that I understand what this means, but the phrasing is very awkward. Figs. 4 and 8 I would suggest briefly mentioning that GSS is multiplied by 2 so that the same scale as for the other metrics can be used.)

to some figures that need to be refined before they may be published

(Fig. 3 The authors should mention somewhere the different intervals used and the fact that the FAR colourmap is inverted

Fig. 9 Why is there no land mask in panel d)?

Fig. 9 If the correlation in panels b) and d) is to be compared to that shown in the previous figures (e.g. Fig. 2) as suggested in the main text, then the colourmaps should be the same.

Fig. 9 Why is the colourbar labelled differently wrt Fig. 1? It would be better if the two were consistent (either numbers or text is fine).

Fig. S3 Depending on how you chose to change the correlation colourmaps in Figs 2 and 9, please ensure that this is consistent too).

2. The results presented in the study are very application-oriented. Why not enhance this aspect by providing the equivalent of Fig. 4 for the operational and optimised forecasts?

3. The reader finds out about the leave-one-out approach only on l. 245. This needs to be discussed before and in more detail, as it is a crucial aspect of the methodology. Related to this, have the authors re-caclulated the WRs every time without the "left out"

[Figure]

year to ensure no information leakage between training and test data?

4. Fig. 1: Why is this only shown for winter if all four seasons are then discussed?

5. ll. 281-287 This is one of the most interesting passages of the paper. Since the authors have clearly performed a comprehensive set of analyses, I would encourage them to expand the discussion of the physical drivers that may be behind these results.

**Other Comments**

1. l. 36 When they say "models" are the authors referring to climate models, NWP models in general, deterministic forecasts, ensembles or what here?

2. l. 53 This repeats what just said above: "WRs are highly teleconnected to ... precipitation" and "They are well known to ... either favour or inhibit precipitation in Europe".

3. It might be helpful to mention the four canoncial weather regimes over the North Atlantic.

4. l. 59 "The WRs also have an impact on extreme events" is a repetition of what said on l. 49 and l. 52.

5. l. 65 As not all readers may be familiar with WRs, the authors should mention that they are often (although not exclusively) diagnosed using 500 hPa geopotential height and maybe provide a reference.

6. l. 91 This is a very odd choice. The authors state that they upscale E-OBS and ENS to 1 degree but then use ERA-Interim at 1.125 degrees (which is certainly not its native resolution). Why not use it at the same resolution as E-OBS (or alternatively at the highest recommended resolution of 0.75 degrees, if it will be at a different resolution from the other data anyways)?

7. ll. 100-103 Would be more logical to have these earlier in the paragraph, before discussing the regridding.

8. l. 129 Are there other studies that recover 3 WRs? If so, cite them. If not, provide a more detailed explanation of why you find a different number. With regards to the possible sources of discrepancy, one may argue that, assuming a more-or-less stationary system, if the number of WRs depends on the period chosen, then the length of the period is simply too short to define WRs in the first place.

9. It would be nice to see the equvalent of Fig. S3 for ENS (even though it is not mentioned in the caption I guess Fig. S3 uses ERA-Interim and E-OBS only?

10. ll. 166-168 Do the authors have some evidence or reference to support this? Do any of the national civil protection services or other public services in Europe routinely make use of this type of information?

11. l. 170 vs l. 108 I am a bit confused as to what is computed up to 2014 and what up to 2013. This is a detail, but if some of the figures/results are indeed computed using 2014 too, then a time interval column could be added to Table 1.

12. l. 233 An improvement with respect to what?

13. l. 279 Is a forecast with such a score useful in an operational context? More generally, can the authors make an assessment of where (geographically speaking) and to what extent their method would actually provide "operationally useful" information? I want to clarify that I am not suggesting the paper would be less valuable if no such operational information can be obtained from the results presented in it. However, I think that an honest discussion of this aspect would make the paper more useful to both the research and public service communities.

---

## Referee Comment (RC2) · Anonymous Referee #2 · 14 Aug 2018

Review of manuscript nhess-2018-199 entitled "On the use of Weather Regimes to forecast meteorological drought over Europe" by C. Lavaysse, J. Vogt, A. Toreti, M. Carrera, and F. Pappenberger

This study proposes and evaluates a novel approach to predict meteorological drought on monthly time scales based on the forecast of large-scale flow patterns. The authors show that in the ECMWF extended forecasting system drought forecasts based on weather regime (WR) occurrence outperform drought forecasts based on direct precipitation forecast in most regions of Europe. Particularly those regions (British Isles, Scandinavia, NE Europe) benefit from WR-based drought forecasts, which show a strong link of drought and the large-scale weather regimes. The linkage of WR and drought as well as its predictability is thoroughly investigated; stratified according to

seasons, and sensitivities to drought intensity and previous drought conditions tested. It is shown that the WR approach has even more benefit for stronger droughts. Furthermore, the linkage is stronger in winter than in summer. Finally it is shown that the forecast captures well the linkage between WR and drought, but has difficulties in correctly representing WR frequencies.

Overall this study presents a very important contribution to research on monthly and sub-seasonal predictability and novel applications of now existing operational NWP data. It thoroughly documents that forecast products based on atmospheric fields that are easier to predict in NWP (e.g. geopotential, temperature) than more complex variables (e.g. precipitation, wind) can be effectively used to predict weather impacts due to the linkage of flow patterns and surface weather. The paper is well organised, clearly written in most parts and the figures carefully designed and chosen to support the storyline. Only at few places I struggled to follow and some references (to tables) were misplaced. Despite the long list of comments (which are all minor) I recommend to accept after one round of revisions.

Broader Comments:

1. Several studies document that predictability on monthly time scales primarily arises from predictability in week 1 and week 2, while it vanishes in week 3-4 (e.g. Ferranti et al. 2018, Vigaud et al 2017, 2018). Did you check weekly or two-weekly forecast skill for the drought events? How would a two-weekly stratification look like?

2. To me the usage of the term "teleconnection" is misleading. I understand under this term large-scale linkages from e.g. the Madden-Julian-Oscillation or SST or ENSO on weather regimes. In this paper I would talk of a linkage between the weather regimes and smaller-scale local weather/precipitation.

3. Please carefully revise and check how you introduce your terminology. Sometimes different terms are used for similar items, some terms are poorly or not introduced. This makes the paper in partly difficult to read. Details are given in the line-by-line

comments.

4. Please also explain some of the methodology in more detail.

5. Some more literature could be cited: E.g. studies by Lavers et al. 2016ab and Ferranti et al. 2018, also support the idea that large-scale fields provide more predictability for a local weather phenomenon than trying to predict the phenomenon itself. Linkage to climate change could be mentioned e.g. with Santos et al. 2016 or Schaller et al. 2018 in the outlook. Linkage of weather regimes to other surface variables e.g. wind could be mentioned (e.g. Grams et al. 2017).

6. Table references are mixed up. Also order these in their order of occurrence in the paper. I found it difficult to directly understand tables and figures, due to too little information in the caption - in particular for tables. All Supplemental Figures should also be cited in the main text in their order of appearance.

Detailed comments:

reference order: Does NHESS require stating the most recent literature first? If not please revert.

l51: refer also to Ferranti et al. 2018 as a recent study on WR and cold extremes. For wind Grams et al. 2017 might be an appropriate reference

l52: avoid talking of being teleconnected -> linked/associated with

l55: up to here you nicely introduced into the WR concept. It becomes confusing (l55-60) to now talk of NAO+-, without having clarified the differences between the NAO and WR concepts and without having clarified that the two NAO phases are two of the 4 winter regimes. So consider to first contrast NAO (as only describing part (i.e. 30%) of the large-scale variability on monthly/seasonal time scales) to WR (as describing most of the variability (i.e. 75%) on monthly time scales).

Section 2b: Do you do the k-means clustering in physical or phase space? E.g. is

an EOF analysis performed? Do you use time-filtering? You should provide few (2-3 sentences) more details on the WR definition and also more details on how individual days are attributed to a WR (e.g. in physical or phase space l130ff). You repeatedly state why you only use WRs based on a k-means clustering in ERAI. Once justifying this approach is sufficient and convincing! Remove the redundancy.

l140/l146: You refer to Table 2 here!? I found it difficult to directly understand Table 2. Re-Reading these lines, I understand it now, but it would help if you repeat the definition of a WR combination in the caption of Table 2. "WR combinations are defined as either additions or subtractions of monthly WR frequencies" Also in Table 2 I would replace WRa, WRb, ... by a, b ....

Table 1 is really helpful but hardly referred in the text. (as Table 6?). Definitely keep it.

l113: season definition appears confusing. Please explicitly state if winter is NDJF or DJF, ...

Section 2c: some more details on how SPI is computed would help pleasae indicate a current (WMO) reference.

l178: should state Table 3? Table 3 is quite standard, needed?

l198: please specify in the text what is meant by "best correlation criterion". Do you compute at each grid point for each of the 16 WR/combination freq. time series the correlation to the SPI-1 time series? Then the WR/combination with highest correlation is shown in Fig. 1? This procedure needs to be stated.

Section 3: clearly introduce the names/terminology used for the different setups/methods presented in Table 1 and subsequently please use it consistently. The definition of the fourth vs. third method remains obscure.

l212: please name this first method as the "Reference" method (as in Table 1) l213: Table 1! l217-219: the sentence in brackets is not needed / redundant. l216: Suggestion for a slightly clearer formulation: "The third forecasting method, called "operational"

in Table 1, computes MOAWR from ensemble data attributed to the WRs based on ERAI (see Section 2b)." l219ff: Here I am confused: Do you still compute WR-k-means clustering based on ENS data, or do you mean, that WR anomalies are computed wrt. ENS climatology not ERA-I climatology.

l228: The term "Reference" is not introduced, yet.

The bulk of the result sections is well-written, therefore I have less detailed comments in the following.

l281: it would be nice to show evidence for these findings on the other seasons in the Supplement.

Section 5a: The WR evaluation section is a bit weak. Please refer to studies by Ferranti et al. 2015, 2017 or Matsueda and Palmer for approaches of WR evaluation. E.g. how are weekly WR freq. evolving with time?

l303: the statement about blocking is vague. Do you refer to the Blocking regime (so one of the 4 WR) or blocking anticyclones in general?

l304: The sentence starting in line 304 and ending in line 309 is complicated. Perhaps directly start with what is shown in Figure 7 then go into the interpretation. I do not understand the argumentation of causes and effect for the anomalies. Try to rewrite lines 298-312 in clearer language. I wonder if Fig. 6 and 7 show really different things, or if only one of the two is sufficient. I prefer Fig. 7 but would elaborate on its description and interpretation.

l311: you mean Table 4. But except here, the Table is hardly used. Do you really need it?

l404: Ferranti et al. 2015, Weisheimer 2016, Magnusson 2017 and/or Grams et al. 2018 could also be cited to highlight the challenges in predicting WR. Furthermore you could insert a statement on the evolution of WR under climate change e.g. Santos et al. 2016, Schaller et al. 2018.

References:

Ferranti, L., L. Magnusson, F. Vitart, and D. S. Richardson, How far in advance can we predict changes in large-scale flow leading to severe cold conditions over Europe? Quarterly Journal of the Royal Meteorological Society, 0, doi:10.1002/qj.3341.

Ferranti, L., S. Corti, and M. Janousek, 2015: Flow-dependent verification of the ECMWF ensemble over the Euro-Atlantic sector. Q.J.R. Meteorol. Soc., 141, 916–924, doi:10.1002/qj.2411.

Grams, C. M., R. Beerli, S. Pfenninger, I. Staffell, and H. Wernli, 2017: Balancing Europe/'s wind-power output through spatial deployment informed by weather regimes. Nature Climate Change, 7, 557–562, doi:10.1038/nclimate3338.

Grams, C. M., L. Magnusson, and E. Madonna, An atmospheric dynamics' perspective on the amplification and propagation of forecast error in numerical weather prediction models: a case study. Quarterly Journal of the Royal Meteorological Society, 0, doi:10.1002/qj.3353.

Lavers, D. A., F. Pappenberger, D. S. Richardson, and E. Zsoter, 2016a: ECMWF Extreme Forecast Index for water vapor transport: A forecast tool for atmospheric rivers and extreme precipitation. Geophys. Res. Lett., 43, 2016GL071320, doi:10.1002/2016GL071320.

Lavers, D. A., D. E. Waliser, F. M. Ralph, and M. D. Dettinger, 2016b: Predictability of horizontal water vapor transport relative to precipitation: Enhancing situational awareness for forecasting western U.S. extreme precipitation and flooding. Geophys. Res. Lett., 43, 2016GL067765, doi:10.1002/2016GL067765.

Magnusson, L., 2017: Diagnostic methods for understanding the origin of forecast errors. Q.J.R. Meteorol. Soc, 143, 2129–2142, doi:10.1002/qj.3072.

Santos, J. A., M. Belo-Pereira, H. Fraga, and J. G. Pinto, 2016: Understanding climate change projections for precipitation over western Europe with a weather typing approach. J. Geophys. Res. Atmos., 121, 2015JD024399, doi:10.1002/2015JD024399.

Schaller, N., J. Sillmann, J. Anstey, E. M. Fischer, C. M. Grams, and S. Russo, 2018: Influence of blocking on Northern European and Western Russian heatwaves in large climate model ensembles. Environ. Res. Lett., 13, 054015, doi:10.1088/1748-9326/aaba55.

Vigaud, N., A. W. Robertson, and M. K. Tippett, 2017: Multimodel Ensembling of Subseasonal Precipitation Forecasts over North America. Mon. Wea. Rev., 145, 3913–3928, doi:10.1175/MWR-D-17-0092.1.

Vigaud, N., A. w. Robertson, and M. K. Tippett, 2018: Predictability of Recurrent Weather Regimes over North America during Winter from Submonthly Reforecasts. Mon. Wea. Rev., 146, 2559–2577, doi:10.1175/MWR-D-18-0058.1.

Weisheimer, A., N. Schaller, C. O'Reilly, D. A. MacLeod, and T. Palmer, 2017: Atmospheric seasonal forecasts of the twentieth century: multi-decadal variability in predictive skill of the winter North Atlantic Oscillation (NAO) and their potential value for extreme event attribution. Q.J.R. Meteorol. Soc, 143, 917–926, doi:10.1002/qj.2976.
* * *

---

## Author Comment (AC1) · 8 Oct 2018

The authors present a novel approach to provide early warnings for drought events over Europe, based on ECMWF's ERA-Interim reanalysis and ENS forecast system. I really like their basic idea, as it combines a scientifically interesting result with a very practical and applications-oriented framework. Furthermore, the authors have clearly performed an extensive set of analyses in support of their results.

However, I think the manuscript falls short of publication requirements in its current form. In addition to some comments of a more scientific nature, my main concern is the poor form of the submission. The text is often grammatically flawed or very unclear; references to tables are incorrect (and in one case to a table that doesn't exist, at least in the version of the paper I reviewed), and some existing tables are never referenced. Overall the manuscript has a very unrefined feel.

I reiterate that I do find the motivation behind the study and the authors' approach of great interest, so I would encourage them to perform a very thorough review such that their results may be published in NHESS.

We thank the reviewer and we apologise for the typos and wrong numbering of tables. We have revised completely the manuscript correcting all those errors. We have replied to all his/her comments in red.

Major Comments
1. As stated above, I deem the manuscript to be in an advanced draft form rather than at a publication-ready stage. The issues range from simple typos/grammatical errors/incorrectly referenced tables/oversights (some examples here:
l. 25 "While, the"
Corrected

l. 29 "and onset drought events"
Corrected

l. 69 "teleconnections in between"
Corrected

ll. 84-85 "observational daily station-based" This is a bit redundant. If something is station based it is very likely to also be observational.
"which provides observational daily station-based precipitation" has been modified by "which provides daily station-based precipitation"

ll. 96 Space between "10" and "days" and "32" and "days". Remove multiple spaces between "Tl319" and "64".
Modified as suggested

l. 97 "1-degree" no need for hyphen.
Modified as suggested

l. 102 "launched" –> "initialised"
Modified as suggested

l. 111 "done, exclusively"
Corrected

l. 121 There is a lone parenthesis
Modified as suggested

l. 121 "previous studies mentioned earlier" –> "aforementioned studies".
Modified as suggested

l. 130 "identify The closest"
Corrected

l. 146 "Table 1". Do the authors here mean Table 2? The real Table 1 actually does not seem to be referenced anywhere in the text.
We are sorry for these misunderstandings due to a mistake in the list of tables. They have been all corrected.

l. 156 "datasets from observations" –> "observational datasets"
Modified as suggested

ll. 157-158 "The choice ... has been verified in Lavaysse et al. (2015) and shown that this assumption"
Modified as suggested

ll. 159-160 "some grid points the significant tests are not verified"
Corrected

l. 178 "Table 2" –> "Table 3"?
Modified. See previous comment

l. 183 "dump" –> "lower"
This has been modified

l.185 "Peirce" –> "Pierce"
This has been corrected

l. 198 "leads to coherent picture"
Modified by "leads to depict a coherent picture"

ll. 201-202 "they are generally higher teleconnections"
This sentence has been removed.

ll. 207-208 "the potential beneifts is assessed"
Corrected

l. 213 There is no table 6 in the version of the paper I have reviewed (indeed the tables stop at 4).
The list and the reference of tables have been corrected.

l. 218 "to avoid potential problem"
We have clarified these problems and replaced by "to avoid major changes in the WR classification"

ll. 227-228 "also called and presented in the previous section as Reference"
We have clarified this point by clearly mentioning that the forecast based on precipitation is called the Reference.

l. 229 "forecasts" –> "forecast"
Corrected as suggested.

l. 247 "anomaly" –> "anomalies"
Corrected as suggested.

l. 261 "2.5l. 269 "for all the domain shown in the previous Figures" –> "for the whole domain shown in the previous figures"
Modified as suggested.

l. 311 Again a reference to Table 6.
Please see previous comment

l. 316 "e.g." or "i.e."?
Modified. We have also checked all the uses of these latin abbreviations to avoid other mistakes.

l. 328 "intensities" –> "magnitudes"?
Modified as suggested

l. 344 "showing a more complex observed than forecasted teleconnections"
"complex" has been replaced by "variable"

l. 346 "the relative good representation"
The sentence has been rephrased and the similarities have been quantified.
The new sentence is now:
"Nevertheless over continental regions, there are similarities between the assigned predictors obtained by using ENS and ERAI (impact of WRs b, b-a, a, c-d), with more than 60% of agreement, illustrating the good representation of the impacts of specific WR on precipitation by ENS."

l. 347 "The correlation between the WRs forecasted and the observed precipitation"
Corrected

l. 385 "the dynamic of precipitation"
Replaced by "the atmospheric dynamic associated with the precipitation"

l. 390 "The skill scores is"
Corrected

l. 405 "with regard to" –> "for"
Corrected

l. 408 Acknowledgements are missing.
The acknowledgements have been added

Tables 3 and 4 are never referred to in the text.
See previous comment

Fig. 8 caption "GSS*2:w"
Corrected

The figures in the SI aren't prefaced by "S" which makes it hard to figure out whether the authors in the SI refer to the SI figures or those in the main text.
We have now clarified the SI and added S when the text refers to a figure in the SI.

l. 72 in the SI: "?" in place of a reference.)
Corrected

to some very unclear or contradictory passages that I would recommend the authors rephrase (some examples here:
ll. 30-32 This is a bit confusing: if the MOAWRs are anomalies of occurrence how can they "depict" a large-scale atmospheric pattern?
We have modified the sentence as follows:
"Finally, in most of the cases, the ENSemble system of the ECMWF successfully represents the observed large scale atmospheric patterns, depicted by the MOAWRs, associated with drought events over Europe." has been replaced by:
"Finally, the results show that the ENSemble system of the ECMWF successfully represents most of the observed linkages between large scale atmospheric patterns, depicted by the WRs, and drought events over Europe."

ll. 48-52 The authors first discuss precipitation and then switch to wind gusts and temperature extremes without any apparent connection. How are the latter two fields relevant to the study?
This part of the sentence has been removed.

ll. 202-203 Why speak of northern and southern Europe and then shift to central and north western Europe?
This sentence has been removed.

ll. 212-224 This passage is very important (the authors explain a key aspect of their approach) but also occasionally difficult to understand. For example, what does "uses ENS for the WRs assignation" mean? Do the authors mean that the WRs are defined using the ENS dataset? Similarly, what does "modelled precipitation" refer to? Do the authors mean ENS precipitation (ERA-Interim is a model too)? Linked to the above, the passage describing the forecasting methodologies in the SI (to which the authors point the reader) never explicitly mentions the "idealized" approach, although this is included in Fig. S1. I would suggest the authors simply

include the full description of the methodologies in the main paper (the text in the SI is not that much longer than that already in the main paper), leaving the details of the attribution and MDA analysis in the SI. Again linked to the above, since the authors have given names to the different forecasting approaches, they should use them! For e.g. the caption in Fig. 4 never names the approach it depicts. This is one example, but there are a number of other similar cases throughout the submission.

According to this comment, we have decided to move the text from the SI to the main document. The description of the methodology has been corrected and simplified. Also the names of each experiment are now well defined and explicitly mentioned during the description of the results.

ll. 228 Actually, this is never called "Reference" in the previous section.

We have now mentioned what the reference experiment is (derived from the forecast of precipitation) in the previous section.

ll. 305-307 The authors first speak about "overestimation of low occurrences in the observations", which is confusing because observations (or rather reanalysis, if I understood what the paragraph talks about) here are taken as the ground truth and so cannot under or overestimate occurrences. Next they mention the "larger number of forecasted events compared to observed ones with durations shorter than 5 days". I take this to mean that the forecasts produce a larger number of short events than the reanalysis. However, from Figure 6 it seems that short events are more frequent in ERA-Interim than in the ENS data. I may have misunderstood the whole passage, but in that case other readers may well have the same issue.

We thank the reviewer for this comment. We have clarified through all the manuscript that WRs (and so MOAWRs) derived from ERAI are not observations.

The two figures (Figs. 6 and 7) and the associated paragraphs are different and discuss about two different results. The first part is about the frequencies of each duration, without any information about the concomitant values between ERAI and ENS. The distributions of long and short durations of each WR are checked. In the second part, the temporal evolution of the MOAWR values provided by ENS and ERAI are then compared and the concomitant values analysed. For these reasons, the conclusions can be different. Nevertheless, we agree that these sentences could be misleading. Therefore, we have clarified the text. The sentences:
"The WR-distributions as given by the forecasts are characterized by a higher degree of similarity than the ones given by ERAI, with a peak of occurrence at around 5-8 days in winter (blue bars, Fig.6). The same holds for the other seasons (not shown). The lower spread of the forecasted WR occurrences, associated with reduced tails (i.e. reduced occurrences for durations exceeding 20 days), could be explained by the underestimation of the long-term blocking. A further examination of the temporal evolution of these occurrence anomalies suggests that the distribution of forecasted drought occurrences (previously shown) could mainly explain the overestimation of low occurrences using the reanalysis (i.e., larger number of forecasted events compared to those derived from ERAI with durations shorter than 5 days) and the underestimation of longer duration events (i.e., lower events with durations longer than 15 days using ENS than ERAI, red dotted lines in Fig. 7)."
have been replaced by :
"The WR-distributions as given by the forecasts are characterized by a higher degree of similarity between the WRs than the ones given by ERAI, with a peak of occurrence at around

5-8 days in winter for the four distributions (blue bars, Fig. 6a-d). The same holds for the other seasons (not shown). The lower spread of the forecasted WR occurrences, associated with reduced tails (i.e. reduced occurrences for durations exceeding 20 days), could be explained by the underestimation of the longer blocking episodes. A further comparison of the MOAWRs from ERAI and ENS (scatter plots in Fig. 7) suggests that : i) the distribution of forecasted drought occurrences could be explained by the overestimation of low occurrences using ENS than the reanalysis (i.e., larger number of forecasted events compared to those derived from ERAI with durations shorter than 5 days) and, ii) the underestimation of longer duration events (i.e., lower events with durations longer than 15 days using ENS than ERAI, red dotted lines in Fig. 7). "

ll. 346-351 From this sentence I understand that the correlation values are low where the teleconnection with precipitation is the strongest, while Figure 9 (and logic) suggests the opposite.
We agree that these sentences were not clear. We have modified them.
"The correlation values, which can be compared to the correlation shown in Fig. 9a, are quite low as a result of the relatively low predictability of the WRs previously discussed and where the teleconnection between WRs and precipitation is the highest (e.g. in southern Norway and the northern part of the U.K.)."
has been modified as follows:
"The correlation values, which can be compared to the correlation shown in Fig. 9a, are quite low as a result of the relatively low predictability of the WRs previously discussed. The values are also sensitive to the strength of the teleconnection between WRs and precipitation (i.e. highest scores in southern Norway and the northern part of the U.K, lowest scores in Central Europe.)."

ll. 361-363 Do the authors mean that the results clearly point to the ability in forecasting the large-scale atmospheric circulation as a factor limiting the skill of this approach?
This conclusion needs to be clarified. We point out that due to the good representation of the relationship between MOAWRs and precipitation in ENS, the use of this predictor is rather limited. Indeed, the purpose of this method is based on the use of more predictable predictor (i.e. WRs) than the predictant (i.e. the precipitation). If the connection between the two is well represented, the two variables have relatively close scores in term of predictability. This could limit the benefit of this approach.
We have modified the text as follows:
"This highlights the overall good representation of the processes linking large scale circulation and precipitation deficits by ENS. So the ENS model succeeds in capturing the impacts of the WR occurrence on the precipitation anomalies as shown in the observations over a large part of Europe. These results could suggest limitations in using such predictors as the lack of skill score could result from a failure in forecasting the large-scale atmospheric circulation (rather than from a misrepresentation of the physical processes)."
modified by:
"Despite some differences observed in southern France and Italy where 'Process' overestimates the large-scale forcing on precipitation (i.e. with stronger correlation with WRs than observed), the patterns obtained when comparing the correlation values between SPI-1 (observed or forecasted) and MOAWRs (from ERAI or ENS) in Figs. S3, S5 and S10 are very similar. This highlights the overall good representation by ENS of the processes linking large scale circulation and local precipitation deficits. So the ENS model succeeds in capturing the

impacts of the WR occurrences on the precipitation anomalies as shown with observations and ERAI over a large part of Europe. These results could suggest limitations in using such predictors as the lack of skill score could result from a failure in forecasting the large-scale atmospheric circulation rather than from a misrepresentation of the physical processes from the large scale forcing to local weather."

ll. 391-393 I think that I understand what this means, but the phrasing is very awkward.
The sentence has been modified as follows.
"Due to the location of the geopotential anomalies and the induced wind fields, or connected with the local effects that reduce the influence of the large scale forcing, the impacts of these MOAWRs on precipitation could be low, as observed in winter over central Europe."
has been modified by:
"Due to the distance of the geopotential anomalies to some target regions, or because of some local effects that could be predominant to the large scale forcing, the impacts of these MOAWRs on precipitation could be low, as observed in winter over central Europe."

Figs. 4 and 8 I would suggest briefly mentioning that GSS is multiplied by 2 so that the same scale as for the other metrics can be used.)
Modified as suggested.

to some figures that need to be refined before they may be published
(Fig. 3 The authors should mention somewhere the different intervals used and the fact that the FAR colourmap is inverted
The caption of Fig. 3 has been modified as suggested.

Fig. 9 Why is there no land mask in panel d)?
The land mask has been applied.

Fig. 9 If the correlation in panels b) and d) is to be compared to that shown in the previous figures (e.g. Fig. 2) as suggested in the main text, then the colour maps should be the same.
Figure 9 has been modified as suggested.

Fig. 9 Why is the colourbar labelled differently wrt Fig. 1? It would be better if the two were consistent (either numbers or text is fine).
Changed as suggested

Fig. S3 Depending on how you chose to change the correlation colour maps in Figs 2 and 9, please ensure that this is consistent too).
Figure S3 is a bit different since it shows the correlation values (positive or negative) whereas Figs. 2 and 9 provide absolute values of correlation to highlight the strength of the connection and to select the best predictor. For these reasons, we prefer to keep different colour scales.

2. The results presented in the study are very application-oriented. Why not enhance this aspect by providing the equivalent of Fig. 4 for the operational and optimised forecasts?
The same figure for the 'Optimized' forecast has been added in SI.
Fig. 4 shows already results provided by the 'Operational' forecast. It highlights the benefits of this forecast in relation to the forecast of precipitation ('Reference'). According to this

misunderstanding and the previous comment on the use of the experiment names, we have clarified the caption.

3. The reader finds out about the leave-one-out approach only on l. 245. This needs to be discussed before and in more detail, as it is a crucial aspect of the methodology. Related to this, have the authors re-calculated the WRs every time without the "left out" year to ensure no information leakage between training and test data?

We do not really see the link between the leave-one-out used for the skill scores and the identification of the WRs. Leaving one year out and re-running the WR identification do not modify the already identified WRs as the procedure is stable and avoid the trap of local minima since k-means has been nested with a genetic algorithm.

4. Fig. 1: Why is this only shown for winter if all four seasons are then discussed?

The automatic attributions for the others seasons have been added in the same figure.

5. ll. 281-287 This is one of the most interesting passages of the paper. Since the authors have clearly performed a comprehensive set of analyses, I would encourage them to expand the discussion of the physical drivers that may be behind these results.

Thank you for this comment. Behind the role played by the physical drivers, it is worth to highlight these results are relative to the 'Reference' experiment. Thus, they are also related to the behavior of the atmospheric model and how it represents extreme events. An exhaustive physical interpretation would require a complete and independent analysis that is out of the scope of this study. Nevertheless, we have now mentioned this point in the perspective as follows:

"Finally, the physical drivers should be analysed in detail to better understand why the predictors are more useful when predicting the most extreme events."

Other Comments
1. l. 36 When they say "models" are the authors referring to climate models, NWP models in general, deterministic forecasts, ensembles or what here?

Clarified as suggested.

2. l. 53 This repeats what just said above: "WRs are highly teleconnected to ... precipitation" and "They are well known to ... either favour or inhibit precipitation in Europe".

This sentence has been modified.

3. It might be helpful to mention the four canoncial weather regimes over the North Atlantic.

Mentioned as suggested

4. l. 59 "The WRs also have an impact on extreme events" is a repetition of what said on l. 49 and l. 52.

This sentence has been removed.

5. l. 65 As not all readers may be familiar with WRs, the authors should mention that they are often (although not exclusively) diagnosed using 500 hPa geopotential height and maybe provide a reference.

Added as suggested.

6. l. 91 This is a very odd choice. The authors state that they upscale E-OBS and ENS to 1 degree but then use ERA-Interim at 1.125 degrees (which is certainly not its native resolution). Why not use it at the same resolution as E-OBS (or alternatively at the highest recommended resolution of 0.75 degrees, if it will be at a different resolution from the other data anyways)?

The ERAI datasets are here used to define WRs and high horizontal resolution is not really needed in this exercise. Many studies dealing with WRs are using even coarser datasets, such as NCEP/NCAR with 2.5 degrees resolution.

E-OBS provides local precipitation datasets, so there is no needs to have common resolutions. There is still a potential and limited impact when comparing the WRs distributions provided by ENS and ERAI. But the other resolution available for ERAI is 0.75 degrees that is still different to the ENS ones.

7. ll. 100-103 Would be more logical to have these earlier in the paragraph, before discussing the regridding.

Displaced as suggested.

8. l. 129 Are there other studies that recover 3 WRs? If so, cite them. If not, provide a more detailed explanation of why you find a different number. With regards to the possible sources of discrepancy, one may argue that, assuming a more-or-less stationary system, if the number of WRs depends on the period chosen, then the length of the period is simply too short to define WRs in the first place.

Choosing the optimal number of clusters is still an open statistical issue, therefore, it is a difficult task in any classifying exercise. Most of the studies have focused only on winter and as stated by the reviewer have identified four weather regimes. It is important to highlight that three of them are associated with preferred latitudinal position of the jet stream, while the last one is related to blocking. Now, our analysis confirms the four regimes in winter while it points to 3 regimes in summer and autumn. This difference clearly shows the seasonal dependence of the regimes, since the system cannot be considered stationary (there is a time dependence) and the period can be considered long enough to detect regimes (as also confirmed by the four ones identified in winter).

9. It would be nice to see the equivalent of Fig. S3 for ENS (even though it is not mentioned in the caption I guess Fig. S3 uses ERA-Interim and E-OBS only?

The caption has been clarified. Yes, in Fig. S3 ERAI and E-OBS are used.

We have now added the equivalent of Fig. S3 for 'Optimized' and 'Process' forecasts in SI (new Fig. S5 and S10). The similarities with S3 are remarquable and discussed in the main document.

10. ll. 166-168 Do the authors have some evidence or reference to support this? Do any of the national civil protection services or other public services in Europe routinely make use of this type of information?

We have added references.

11. l. 170 vs l. 108 I am a bit confused as to what is computed up to 2014 and what up to 2013. This is a detail, but if some of the figures/results are indeed computed using 2014 too, then a time interval column could be added to Table 1.

All the data are computed up to 2013. 2014 is the end of the forecast period, but we used only the hindcasts of these forecasts to get significant results without big changes of the model

versions (as mentioned in the datasets section). So all the dataset in Table are going up to 2013.

12. l. 233 An improvement with respect to what?
'... an improvement ...' has been modified by '... high scores ...'

13. l. 279 Is a forecast with such a score useful in an operational context? More generally, can the authors make an assessment of where (geographically speaking) and to what extent their method would actually provide "operationally useful" information?
I want to clarify that I am not suggesting the paper would be less valuable if no such operational information can be obtained from the results presented in it. However, I think that an honest discussion of this aspect would make the paper more useful to both the research and public service communities.
We agree with adding discussion about that. Nevertheless, we cannot evaluate the usefulness of this operational forecast without taking into account the costs of the damages in each case of the contingency table (hits, misses, false alarms). Moreover it is quite difficult to find a common and general cost function for all the users. These costs may strongly vary depending on their applications (i.e. civil protection, water management services, farmers ...).
The only statement we can provide is that, according to the probabilistic scores, there is a significant improvement in using forecasts w.r.t. the climatology (based on the GSS). We have also shown that the forecasts using predictors generate, in some regions and in some seasons, significant improvements.
Most of the weather services provide now forecasts up to several months. It seems very important to honestly evaluate the added values of these forecasts. That is the main objective of this study.
This paragraph has been added in the conclusion:
"Most of the weather services provide now forecasts up to several months. For users, it appears essential to scientifically and statistically evaluate the added values of these forecasts for specific extreme events such as meteorological droughts. This is the main objective of this study.
Nevertheless, evaluating the practical usefulness of this operational forecast is difficult without taking into account the costs for each case of the contingency table (hits, misses, false alarms) that strongly vary depending on their applications (i.e. civil protection, water management services, farmers' decision supporting systems, etc.). The statement provided in this study is based on statistical scores independent of these costs. According to the GSS, there is a significant improvement of using forecasts in relation to the climatology. Moreover, the forecasts using predictors generate, in some regions and some seasons, significant improvements of these forecasts by using the same score. To evaluate these improvement for specific users, the costs should be taken into account that is a major perspective of this study."

---

## Author Comment (AC2) · 8 Oct 2018

Review of manuscript nhess-2018-199 entitled "On the use of Weather Regimes to forecast meteorological drought over Europe" by C. Lavaysse, J. Vogt, A. Toreti, M. Carrera, and F. Pappenberger

This study proposes and evaluates a novel approach to predict meteorological drought on monthly time scales based on the forecast of large-scale flow patterns. The authors show that in the ECMWF extended forecasting system drought forecasts based on weather regime (WR) occurrence outperform drought forecasts based on direct precipitation forecast in most regions of Europe. Particularly those regions (British Isles, Scandinavia, NE Europe) benefit from WR-based drought forecasts, which show a strong link of drought and the large-scale weather regimes. The linkage of WR and drought as well as its predictability is thoroughly investigated; stratified according to seasons, and sensitivities to drought intensity and previous drought conditions tested.

It is shown that the WR approach has even more benefit for stronger droughts. Furthermore, the linkage is stronger in winter than in summer. Finally it is shown that the forecast captures well the linkage between WR and drought, but has difficulties in correctly representing WR frequencies.

Overall this study presents a very important contribution to research on monthly and sub-seasonal predictability and novel applications of now existing operational NWP data. It thoroughly documents that forecast products based on atmospheric fields that are easier to predict in NWP (e.g. geopotential, temperature) than more complex variables (e.g. precipitation, wind) can be effectively used to predict weather impacts due to the linkage of flow patterns and surface weather. The paper is well organised, clearly written in most parts and the figures carefully designed and chosen to support the storyline. Only at few places I struggled to follow and some references (to tables) were misplaced. Despite the long list of comments (which are all minor) I recommend to accept after one round of revisions.

First, we would like to thank the reviewer for his/her positive and encouraging response highlighting the importance of this scientific paper. We have replied to all his/her comments in red.

Broader Comments:

1. Several studies document that predictability on monthly time scales primarily arises from predictability in week 1 and week 2, while it vanishes in week 3-4 (e.g. Ferranti et al. 2018, Vigaud et al 2017, 2018). Did you check weekly or two-weekly forecast skill for the drought events? How would a two-weekly stratification look like?

Thanks for the references which point to an important topic. We focus on specific drought duration that could be relevant for users and decision makers (WMO No.1090). The scores shown in the study are obviously related to the ones at weekly or two-weekly time scales and skill is influenced by the performance of these initial weeks. However, we have not quantified this contribution as the objective of this study was the assessment of specific drought events on monthly time scale. As this topic deserves further investigation, we plan a dedicated study on it.

2. To me the usage of the term "teleconnection" is misleading. I understand under this term large-scale linkages from e.g. the Madden-Julian-Oscillation or SST or ENSO on weather regimes. In this paper I would talk of a linkage between the weather regimes and smaller-scale local weather/precipitation.

We agree with the reviewer.
As suggested, 'teleconnection' has been replaced by 'linkage' throughout the document.

3. Please carefully revise and check how you introduce your terminology. Sometimes different terms are used for similar items, some terms are poorly or not introduced. This makes the paper in partly difficult to read. Details are given in the line-by-line comments.

Thank you for this helpful comment. Please see the responses of the line-by-line comments.

4. Please also explain some of the methodology in more detail.

As requested, we have substantially modified the explanation of the methodology. All the steps have been displaced from SI to the main document, and we have clarified the entire section.

5. Some more literature could be cited: E.g. studies by Lavers et al. 2016ab and Ferranti et al. 2018, also support the idea that large-scale fields provide more predictability for a local weather phenomenon than trying to predict the phenomenon itself. Linkage to climate change could be mentioned e.g. with Santos et al. 2016 or Schaller et al. 2018 in the outlook. Linkage of weather regimes to other surface variables e.g. wind could be mentioned (e.g. Grams et al. 2017).

Thank you for these references. Some of them have been added as suggested. Nevertheless, according to the comments of Reviewer 1, the impacts of WRs onto other surface variables have been removed and because the link with climate studies is not straightforward, we prefer to not discuss that.

6. Table references are mixed up. Also order these in their order of occurrence in the paper. I found it difficult to directly understand tables and figures, due to too little information in the caption - in particular for tables. All Supplemental Figures should also be cited in the main text in their order of appearance.

We are sorry about the mistakes with the order of tables and references. All the figures are now cited in the main document in good order. We have also improved the captions of some figures/table for clarification.

Detailed comments:
reference order: Does NHESS require stating the most recent literature first? If not please revert.

Done as suggested.

l51: refer also to Ferranti et al. 2018 as a recent study on WR and cold extremes. For wind Grams et al. 2017 might be an appropriate reference

Thank you for providing these interesting references. But following the comment of reviewer #1, the impacts of WRs on other variables have been removed.

l52: avoid talking of being teleconnected -> linked/associated with

See previous comment

l55: up to here you nicely introduced into the WR concept. It becomes confusing (l55-60) to now talk of NAO+-, without having clarified the differences between the NAO and WR concepts and without having clarified that the two NAO phases are two of the 4 winter regimes. So consider to first contrast NAO (as only describing part (i.e. 30%) of the large-scale variability on monthly/seasonal time scales) to WR (as describing most of the variability (i.e. 75%) on monthly time scales).

To clarify this point, a sentence has been added
"First and principally studied in Winter time, when they are more stronger, 4 main states have been defined namely the positive North Atlantic Oscillation phases (NAO+), the negative NAO (NAO-), the Blocking and the Atlantic Ridge."

Section 2b: Do you do the k-means clustering in physical or phase space? E.g. is an EOF analysis performed? Do you use time-filtering? You should provide few (2-3 sentences) more details on the WR definition and also more details on how individual days are attributed to a WR (e.g. in physical or phase space l130ff). You repeatedly state why you only use WRs based on a k-means clustering in ERAI. Once justifying this approach is sufficient and convincing! Remove the redundancy.
The GA-K means (Genetic Algorithm k-means) has been performed on the anomalies of geopotential height at 500 hPa while the attribution for the other cases has been done by minimizing the distance from the centroid. Redundancies have been removed and the methodology section deeply clarified.

l140/l146: You refer to Table 2 here!? I found it difficult to directly understand Table 2. Re-Reading these lines, I understand it now, but it would help if you repeat the definition of a WR combination in the caption of Table 2. "WR combinations are defined as either additions or subtractions of monthly WR frequencies" Also in Table 2 I would replace WRa, WRb, ... by a, b ....
We apologise once more for all the mistakes on tables numbering that have now been corrected. We suppose the reviewer refers to Table 1. As suggested, we have modified the caption and the names of the WRs.

Table 1 is really helpful but hardly referred in the text. (as Table 6?). Definitely keep it.
See previous comment.

l113: season definition appears confusing. Please explicitly state if winter is NDJF or DJF, ...
As mentioned already in the text, winter is defined from December to February.

Section 2c: some more details on how SPI is computed would help pleasae indicate a current (WMO) reference.
The WMO reference has been added. As this method is really common we consider brief description as sufficient enough. Nevertheless, we have clarified some descriptions.

l178: should state Table 3? Table 3 is quite standard, needed?
The references have been corrected. Because all the scores are based on the different values of the contingency table, we prefer to keep it.

l198: please specify in the text what is meant by "best correlation criterion". Do you compute at each grid point for each of the 16 WR/combination freq. time series the correlation to the SPI-1 time series? Then the WR/combination with highest correlation is shown in Fig. 1? This procedure needs to be stated.
This section has been completely reorganized and clarified according to previous comments and comments from reviewer #1

Section 3: clearly introduce the names/terminology used for the different setups/methods presented in Table 1 and subsequently please use it consistently. The definition of the fourth vs. third method remains obscure.

We use now the names of the different experiments introduced in table 1.
The differences between experiment 3 and 4 are better explained (see previous comment).

l212: please name this first method as the "Reference" method (as in Table 1)
Please see previous comment.

l213: Table 1!
Please see previous comment.

l217-219: the sentence in brackets is not needed / redundant.
This sentence has been modified (see the following comment).

l216: Suggestion for a slightly clearer formulation: "The third forecasting method, called "operational" in Table 1, computes MOAWR from ensemble data attributed to the WRs based on ERAI (see Section 2b)."
Modified as suggested.

l219ff: Here I am confused: Do you still compute WR-k-means clustering based on ENS data, or do you mean, that WR anomalies are computed wrt. ENS climatology not ERA-I climatology.
As mentionned previously, this section has been deeply clarified. The WR-k-means clustering is, for all the experiments, done by using ERA-I climatology. ENS is used during the attribution phase to define which WR is forecasted.

l228: The term "Reference" is not introduced, yet.
Reference is now clearly introduced before this paragraph.

The bulk of the result sections is well-written, therefore I have less detailed comments in the following.
l281: it would be nice to show evidence for these findings on the other seasons in the Supplement.
The results for the others seasons have been added in SI

Section 5a: The WR evaluation section is a bit weak. Please refer to studies by Ferranti et al. 2015, 2017 or Matsueda and Palmer for approaches of WR evaluation. E.g. how are weekly WR freq. evolving with time?
Thank you for the references, that have been added and discussed in the section.
The purpose of this study is not to deeply evaluate the forecasted WRs. Only the errors that could influence the results, and so could be considered as source of uncertainties of the drought forecasts, are analysed. That is why this evaluation is not complete but adapted to that specific discussion section.

l303: the statement about blocking is vague. Do you refer to the Blocking regime (so one of the 4 WR) or blocking anticyclones in general?
Here we refer to the Blocking regime. This has been clarified in the text.

l304: The sentence starting in line 304 and ending in line 309 is complicated. Perhaps directly start with what is shown in Figure 7 then go into the interpretation. I do not understand the argumentation of causes and effect for the anomalies. Try to rewrite lines 298-312 in clearer language. I wonder if Fig. 6 and 7 show really different things, or if only one of the two is sufficient. I prefer Fig. 7 but would elaborate on its description and interpretation.

According to this comment and the comment from reviewer 1, we have clarified the text. These sentences:

"The WR-distributions as given by the forecasts are characterized by a higher degree of similarity than the ones given by ERAI, with a peak of occurrence at around 5-8 days in winter (blue bars, Fig.6). The same holds for the other seasons (not shown). The lower spread of the forecasted WR occurrences, associated with reduced tails (i.e. reduced occurrences for durations exceeding 20 days), could be explained by the underestimation of the long-term blocking. A further examination of the temporal evolution of these occurrence anomalies suggests that the distribution of forecasted drought occurrences (previously shown) could mainly explain the overestimation of low occurrences using the reanalysis (i.e., larger number of forecasted events compared to those derived from ERAI with durations shorter than 5 days) and the underestimation of longer duration events (i.e., lower events with durations longer than 15 days using ENS than ERAI, red dotted lines in Fig. 7)."

have been replaced by :

"The WR-distributions as given by the forecasts are characterized by a higher degree of similarity between the WRs than the ones given by ERAI, with a peak of occurrence at around 5-8 days in winter for the four distributions (blue bars, Fig. 6a-d). The same holds for the other seasons (not shown). The lower spread of the forecasted WR occurrences, associated with reduced tails (i.e. reduced occurrences for durations exceeding 20 days), could be explained by the underestimation of the longer blocking episodes. A further comparison of the MOAWRs from ERAI and ENS (scatter plots in Fig. 7) suggests that : i) the distribution of forecasted drought occurrences could be explained by the overestimation of low occurrences using ENS than the reanalysis (i.e., larger number of forecasted events compared to those derived from ERAI with durations shorter than 5 days) and, ii) the underestimation of longer duration events (i.e., lower events with durations longer than 15 days using ENS than ERAI, red dotted lines in Fig. 7). "

l311: you mean Table 4. But except here, the Table is hardly used. Do you really need it?

Please see previous comment.

l404: Ferranti et al. 2015, Weisheimer 2016, Magnusson 2017 and/or Grams et al. 2018 could also be cited to highlight the challenges in predicting WR. Furthermore you could insert a statement on the evolution of WR under climate change e.g. Santos et al. 2016, Schaller et al. 2018.

Thank you for these references that have been added in the manuscript.

As mentioned earlier, the link between this study to those on climate change is not straightforward. Thus, we prefer to not mention this statement.

References:

Ferranti, L., L. Magnusson, F. Vitart, and D. S. Richardson, How far in advance can we predict changes in large-scale flow leading to severe cold conditions over Europe? Quarterly Journal of the Royal Meteorological Society, 0, doi:10.1002/qj.3341.

Ferranti, L., S. Corti, and M. Janousek, 2015: Flow-dependent verification of the ECMWF ensemble over the Euro-Atlantic sector. Q.J.R. Meteorol. Soc., 141, 916–924, doi:10.1002/qj.2411.

Grams, C. M., R. Beerli, S. Pfenninger, I. Staffell, and H. Wernli, 2017: Balancing Europe/'s wind-power output through spatial deployment informed by weather regimes. Nature Climate Change, 7, 557–562, doi:10.1038/nclimate3338.

Grams, C. M., L. Magnusson, and E. Madonna, An atmospheric dynamics' perspective on the amplification and propagation of forecast error in numerical weather prediction models: a case study. Quarterly Journal of the Royal Meteorological Society, 0, doi:10.1002/qj.3353.

Lavers, D. A., F. Pappenberger, D. S. Richardson, and E. Zsoter, 2016a: ECMWF Extreme Forecast Index for water vapor transport: A forecast tool for atmospheric rivers and extreme precipitation. Geophys. Res. Lett., 43, 2016GL071320, doi:10.1002/2016GL071320.

Lavers, D. A., D. E. Waliser, F. M. Ralph, and M. D. Dettinger, 2016b: Predictability of horizontal water vapor transport relative to precipitation: Enhancing situational awareness for forecasting western U.S. extreme precipitation and flooding. Geophys. Res. Lett., 43, 2016GL067765, doi:10.1002/2016GL067765.

Magnusson, L., 2017: Diagnostic methods for understanding the origin of forecast errors. Q.J.R. Meteorol. Soc, 143, 2129–2142, doi:10.1002/qj.3072.

Santos, J. A., M. Belo-Pereira, H. Fraga, and J. G. Pinto, 2016: Understanding climate change projections for precipitation over western Europe with a weather typing approach. J. Geophys. Res. Atmos., 121, 2015JD024399, doi:10.1002/2015JD024399.

Schaller, N., J. Sillmann, J. Anstey, E. M. Fischer, C. M. Grams, and S. Russo, 2018: Influence of blocking on Northern European and Western Russian heatwaves in large climate model ensembles. Environ. Res. Lett., 13, 054015, doi:10.1088/1748-9326/aaba55.

Vigaud, N., A. W. Robertson, and M. K. Tippett, 2017: Multimodel Ensembling of Sub-seasonal Precipitation Forecasts over North America. Mon. Wea. Rev., 145, 3913–3928, doi:10.1175/MWR-D-17-0092.1.

Vigaud, N., A. w. Robertson, and M. K. Tippett, 2018: Predictability of Recurrent Weather Regimes over North America during Winter from Submonthly Reforecasts. Mon. Wea. Rev., 146, 2559–2577, doi:10.1175/MWR-D-18-0058.1.

Weisheimer, A., N. Schaller, C. O'Reilly, D. A. MacLeod, and T. Palmer, 2017: Atmospheric seasonal forecasts of the twentieth century: multi-decadal variability in predictive skill of the winter North Atlantic Oscillation (NAO) and their potential value for extreme event attribution. Q.J.R. Meteorol. Soc, 143, 917–926, doi:10.1002/qj.2976.

---

## Author Response (AR2)

Report #1

This is my second review of the study. The paper has been improved through a clearer writing style, more thorough introduction of the data and methods used, corrected references to figures and tables, as well as improved figure/table captions. Most of my previous comments have been addressed adequately. Still I think my previous broader comment 5 should be adressed more thoroughly. I do not agree, that the minor comment of R1 justifies not adressing this topic in the introduction at all. Still the paper could be stronger if it directly referred to the other work in the context of weather regime surface weather linkages and weather regimes and climate change. This would emphasise the broader context of your work.

Overall the paper is now suitable for publication in NHESS.

We would like to thank the reviewer for the positive comments. Please see below, the point by point replies.

Comments:

Former Broader Comment 5. "Some more literature could be cited: E.g. studies by Lavers et al. 2016ab and Ferranti et al. 2018, also support the idea that large-scale fields provide more predictability for a local weather phenomenon than trying to predict the phenomenon itself. Linkage to climate change could be mentioned e.g. with Santos et al. 2016 or Schaller et al. 2018 in the outlook. Linkage of weather regimes to other surface variables e.g. wind could be mentioned (e.g. Grams et al. 2017)."

That was not in red in the previous version but most of the citations have been added as suggested. Please see p2l2. For climatological purpose, the link is not straightforward, and some differences exist that is why we prefer to not cite the references related to climate purposes.

Still I think there is room on p2 l16ff to mention the robustness of the linkage of hot&dry extremes in climate models (Schaller et al. 2018). On p2 l 13 wind modulation by regimes could also briefly be mentioned. Alternately and in addition the Conclusions might be the place where to discuss potential impact of climate change.

Added as suggested

Actual Supplemental Figure captions still have numbers without the "S" (Fig. 1 instead of Fig. S1). I noticed, that you still refer to these using, the "S" convention, which really helps.

We have to see with the editor if adding this is possible according to the latex template of the journal.

Report #2

The authors have extensively addressed my initial concerns. The new format of Section 3 makes the description of the methodology easier to understand, and the more unclear sections of the study have been improved throughout.

We would like to thank the reviewer for the positive comments.

My only remaining concern before the manuscript may be publication-ready are a number of odd turns of phrase (e.g. "On extreme events" in the introduction, "provide now forecasts" in Section 6 etc.), which I would encourage the authors to address.

Modified as suggested

A further small detail: the colourbar labels of Figs. S5 and S10 are cut in the PDF.

Modified as suggested